# Escaping the Sample Trap: Fast and Accurate Epistemic Uncertainty Estimation with Pairwise-Distance Estimators

## Abstract

In machine learning, the ability to assess uncertainty in model predictions is crucial for decision-making, safety-critical applications, and model generalizability. This work introduces a novel approach for epistemic uncertainty estimation for ensemble models using pairwise-distance estimators (PaiDEs). These estimators utilize the pairwise-distance between model components to establish bounds on entropy, which are then used as estimates for information-based criterion. Unlike recent deep learning methods for epistemic uncertainty estimation, which rely on sample-based Monte Carlo estimators, PaiDEs are able to estimate epistemic uncertainty up to 100 times faster, over a larger input space (up to 100 times) and perform more accurately in higher dimensions. To validate our approach, we conducted a series of experiments commonly used to evaluate epistemic uncertainty estimation: 1D sinusoidal data, *Pendulum-v0*, *Hopper-v2*, *Ant-v2* and *Humanoid-v2*. For each experimental setting, an Active Learning framework was applied to demonstrate the advantages of PaiDEs for epistemic uncertainty estimation.

## 1 Introduction

In this paper, we propose Pairwise-Distance Estimators (PaiDEs) as a non-sample based alternative for estimating epistemic uncertainty in deep ensembles with probabilistic outputs. Epistemic uncertainty, often distinguished from aleatoric uncertainty, pertains to model ignorance and can be reduced by increasing the amount of data available (Hora, 1996; Der Kiureghian & Ditlevsen, 2009; Hüllermeier & Waegeman, 2021). Traditionally, in multi-dimensional regression tasks, epistemic uncertainty has been estimated using Monte Carlo (MC) methods because closed-form expressions are generally lacking in most modeling scenarios (Depeweg et al., 2018; Berry & Meger, 2023). However, as the number of dimensions increases, these MC methods become increasingly reliant on a large number of samples.

PaiDEs offer a non-sample based alternative for estimating information-based criterion in ensemble models with probabilistic outputs (Kolchinsky & Tracey, 2017; Kulak & Calinon, 2021; Kulak et al., 2021). Ensembles can be conceptualized as committees, with each ensemble component serving as a committee member (Rokach, 2010). PaiDEs can synthesize the consensus amongst committee members by calculating the distributional distance between each pair of committee members. Distributional distance is a measure of the distance between two probability distributions. These pairwise-distances are aggregated in a way that accurately estimates the differential entropy of the entire ensemble. Assuming that the pairwise distances can be efficiently calculated, PaiDEs provide an efficient way to estimate epistemic uncertainty that is not sample-dependent.

In this study, we showcase the application of PaiDEs for epistemic uncertainty estimation for ensembles with probabilistic outputs, specifically Normalizing Flows (NFs). Prior research has demonstrated the effectiveness of NFs in capturing heteroscedastic and multi-modal aleatoric uncertainty (Kingma & Dhariwal, 2018; Rezende & Mohamed, 2015). In the context of robotic systems, these characteristics are particularly relevant as robots frequently encounter nonlinear stochastic dynamics. We evaluate our method on an array of regression tasks on robotic datasets in the context of active learning. Our contributions are as follows:

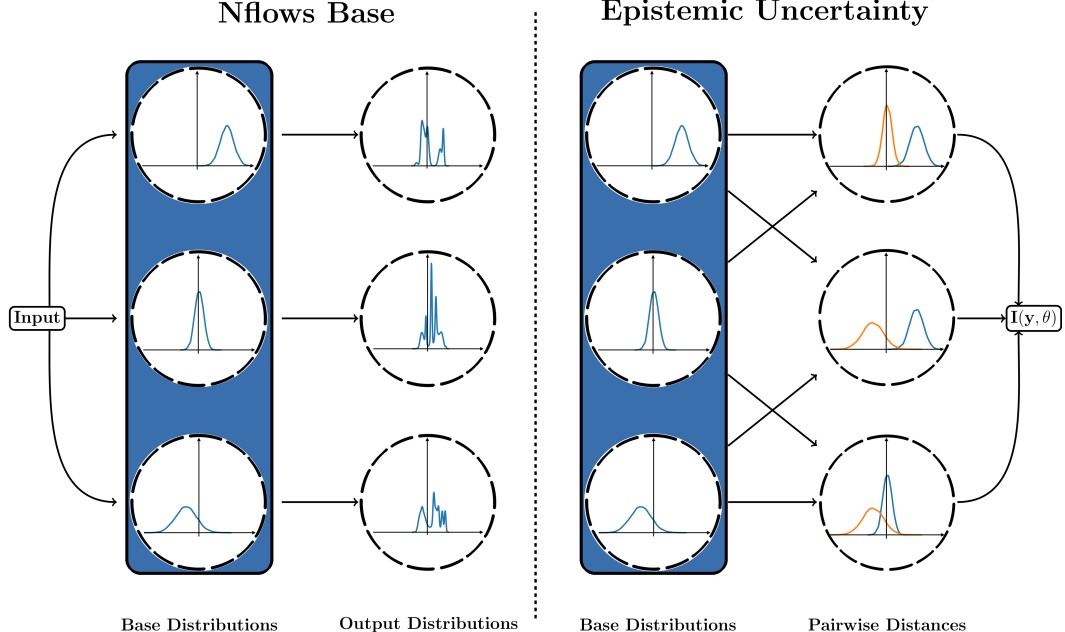

Figure 1: Nflows Base as an ensemble of 3 components with one bijective transformation on the left and an example of the pairwise comparisons needed to estimate epistemic uncertainty for said model on the right. Note the base distributions from Nflows Base are used to estimate the epistemic uncertainty which are highlighted by the blue bar.

- We establish the framework for the application of PaiDEs in the context of estimating epistemic uncertainty for deep ensembles with probabilistic outputs (Section 6).
- We extend previous epistemic uncertainty estimation methods from 11 to 257 dimensions, and demonstrate how PaiDEs outperform MC methods in the higher dimensional setting with rigorous statistical testing (Section 7).
- We provide an analysis of the time saving advantages offered by PaiDEs compared to MC estimators for epistemic uncertainty estimation (Section 7.4).

## 2 PROBLEM STATEMENT

This section provides an overview of the problem at hand. Following a supervised learning framework, let $\mathcal{D} = \{x_i, y_i\}_{i=1}^N$ denote a dataset, where $x_i \in \mathbb{R}^K$ and $y_i \in \mathbb{R}^D$, and our objective is to approximate the conditional probability $p(y|x)$. Let $f_\theta(y, x)$ denote our approximation to the conditional probability density, where $\theta$ is a set of parameters to be learned. The ground-truth distribution, $p(y|x)$, is assumed to take any form including complex multi-modal distributions.

To enable our methods to capture epistemic uncertainty, in addition to complex multi-modal aleatoric uncertainty, we employ ensembles. Ensembles leverage multiple models to obtain the estimated conditional probability by weighting the result output from each ensemble component,

$$f_\theta(y, x) = \sum_{j=1}^M \pi_j f_{\theta_j}(y, x) \qquad \sum_{j=1}^M \pi_j = 1, \tag{1}$$

where $M$ and $0 \le \pi_j \le 1$ are the number of model components and the component weights, respectively. In order to create an ensemble, one of two ways is typically chosen: randomization (Breiman, 2001) or boosting (Freund & Schapire, 1997). While boosting has led to widely used machine learning methods (Chen & Guestrin, 2016), randomization has been the preferred method in deep learning due to its tractability and ease of implementation (Lakshminarayanan et al., 2017).

## 3 EPISTEMIC UNCERTAINTY

Uncertainty is grounded in probability theory and is often analyzed from this perspective (Cover & Thomas, 2006; Hüllermeier & Waegeman, 2021). When capturing uncertainty in supervised learning, one common measure is conditional differential entropy,

$$H(y|x) = -\int p(y|x) \ln p(y|x) dy.$$

Utilizing conditional differential entropy, we can establish an estimate for epistemic uncertainty as introduced by Houlsby et al. (2011), expressed as:

$$I(y, \theta|x) = H(y|x) - E_{p(\theta)}\left[H(y|x, \theta)\right], \tag{2}$$

where $I(\cdot)$ refers to mutual information and $\theta \sim p(\theta)$. Equation (2) demonstrates that epistemic uncertainty, $I(y, \theta|x)$, can be represented by the difference between total uncertainty, $H(y|x)$, and aleatoric uncertainty, $E_{p(\theta)}\left[H(y|x, \theta)\right]$. Mutual information measures the information gained about one variable by observing another. When all components produce the same $f_{\theta_i}(y, x)$, $I(y, \theta|x)$ is zero, indicating no epistemic uncertainty. Conversely, when the components have non-overlapping supports, epistemic uncertainty is high.

Epistemic uncertainty is valuable in decision-making, particularly in active learning (MacKay, 1992; Settles, 2009). We select data points that maximize Equation (2) as in Bayesian Active Learning by Disagreement (BALD) to improve the model's performance (Houlsby et al., 2011). It's worth noting that in the realm of continuous outputs and ensemble models, Equation (2) often lacks a closed-form solution, primarily because total entropy cannot be expressed in closed form,

$$H(y|x) = \int_D \sum_{j=1}^M \pi_j f_{\theta_j}(y, x) \ln \sum_{j=1}^M \pi_j f_{\theta_j}(y, x) dy.$$

Hence, prior methods have resorted to Monte Carlo (MC) estimators for the estimation of epistemic uncertainty (Depeweg et al., 2018; Postels et al., 2020). The Monte Carlo method samples $K$ points from our model, $y_j \sim f_\theta(y, x)$, and then estimates the total uncertainty,

$$\hat{H}_{MC}(y|x) = \frac{-1}{K} \sum_{j=1}^K \ln f_\theta(y_j, x).$$

MC estimators are convenient for estimating quantities through random sampling and are more apt for high-dimensional integrals compared to other numerical methods. However, as the number dimensions increase, MC methods typically require a greater number of samples (Rubinstein & Glynn, 2009).

## 4 PAIRWISE-DISTANCE ESTIMATORS

Unlike MC methods, PaiDEs completely remove this dependence on sampling by leveraging (generalized) distance functions between model component distributions. They can be applied when estimating entropy of mixture distributions as long as the pairwise-distances have a closed-form. Their derivation and properties follow from Kolchinsky & Tracey (2017); we are extending the use of PaiDEs to a supervised learning problem and epistemic uncertainty estimation.

### 4.1 PROPERTIES OF ENTROPY

One can treat a mixture model as a two step process: first a component is drawn and, second, a sample is taken from the corresponding component. Let $p(y, \theta|x)$ denote the joint of our output and model components given input $x$,

$$p(y, \theta|x) = p(\theta_j|x)p(y|\theta_j, x) = \pi_{\theta_j} p(y|\theta_j, x).$$

Now that we have a representation of the joint, following principles of information theory (Cover & Thomas, 2006), we can write its entropy as,

$$H(y, \theta|x) = H(\theta|y, x) + H(y|x). \tag{3}$$

Additionally, one can show the following bounds for $H(y|x)$,

$$H(y|\theta, x) \leq H(y|x) \leq H(y, \theta|x). \tag{4}$$

Intuitively, the lower bound can be justified by the fact that conditioning on more variables can only decrease or keep entropy the same and the upper bound follows from Equation (3) and $H(\theta|y, x) \geq 0$.

## 4.2 PaiDEs Definition

Let $D(p_i \parallel p_j)$ denote a (generalized) distance function between the probability distributions $p_i$ and $p_j$, which for our case represent $p_i = p(y|x, \theta_i)$ and $p_j = p(y|x, \theta_j)$, respectively. More specifically, $D$ is referred to as a premetric, $D(p_i \parallel p_j) \geq 0$ and $D(p_i \parallel p_j) = 0$ if $p_i = p_j$. The distance function need not be symmetric nor obey the triangle inequality. As such, PaiDEs can be defined as,

$$\hat{H}_\rho(y|x) = H(y|\theta, x) - \sum_{i=1}^{M} \pi_i \ln \sum_{j=1}^{M} \pi_j \exp\left(-D(p_i \parallel p_j)\right). \tag{5}$$

PaiDEs have many options for $D(p_i \parallel p_j)$ (Kullback-Leibler divergence, Wasserstein distance, Bhattacharyya distance, Chernoff $\alpha$-divergence, Hellinger distance, etc.).

**Theorem 4.1.** *Using the extreme distance functions,*

$$D_{min}(p_i \parallel p_j) = 0 \quad \forall i, j$$

$$D_{max}(p_i \parallel p_j) = \begin{cases} 0, & if\ p_i = p_j, \\ \infty, & o/w, \end{cases}$$

*one can show that PaiDEs lie within bounds for entropy established in Equation (4).*

Refer to Kolchinsky & Tracey (2017) for the proof. This provides a general class of estimators but a distance function still needs to be chosen. Certain distance functions improve the bounds in Equation (4) and we will use them to guide our choice.

## 4.3 Improved Bounds for PaiDEs

Let the Chernoff $\alpha$-divergence be defined as (Nielsen, 2011),

$$C_\alpha(p_i \parallel p_j) = -\ln \int p^\alpha(y|x, \theta_i) p^{1-\alpha}(y|x, \theta_j) dx,$$

where $\alpha \in [0, 1]$.

**Corollary 4.2.** *When applying Chernoff $\alpha$-divergence as our distance function in Equation (5), we achieve a tighter lower bound than $H(y|\theta, x)$,*

$$\hat{H}_{C_\alpha}(y|x) = H(y|\theta, x) - \sum_{i=1}^{M} \pi_i \ln \sum_{j=1}^{M} \pi_j \exp\left(-C_\alpha(p_i \parallel p_j)\right), \tag{6}$$

$$H(y|\theta, x) \leq \hat{H}_{C_\alpha}(y|x) \leq H(y|x). \tag{7}$$

Refer to Kolchinsky & Tracey (2017) for the proof. In addition, the tightest lower bound can be shown to be $\alpha = 0.5$ for certain situations (Kolchinsky & Tracey, 2017). Note that when $\alpha = 0.5$, the Chernoff $\alpha$-divergence is known as the Bhattacharyya distance,

$$D_B(p_i \parallel p_j) = -\ln \int \sqrt{p(y|x, \theta_i) p(y|x, \theta_j)} dx. \tag{8}$$

We utilized PaiDEs with the Bhattacharyya distance, $\hat{H}_{Bhatt}(y|x) = \hat{H}_{C_{0.5}}(y|x)$, as one proposed improvement to MC estimators.

In addition to the improved lower bound, there is an improved upper bound as well. Let Kullback-Liebler (KL) divergence be defined as follows,

$$D_{KL}(p_i \parallel p_j) = \int p(y|x, \theta_i) \ln \frac{p(y|x, \theta_i)}{p(y|x, \theta_j)} dx.$$

Note that the KL divergence does not satisfy the triangle inequality nor is it symmetric, thus it is not metric but does suffice as a (generalized) distance function.

**Corollary 4.3.** *When applying Kullback-Liebler divergence as our distance function in Equation (5), we achieve a tighter upper bound than $H(y, \theta|x)$,*

$$\hat{H}_{KL}(y|x) = H(y|\theta, x) - \sum_{i=1}^{M} \pi_i \ln \sum_{j=1}^{M} \pi_j \exp\left(-D_{KL}(p_i \parallel p_j)\right), \tag{9}$$

$$H(y|x) \leq \hat{H}_{KL}(y|x) \leq H(y, \theta|x). \tag{10}$$

Refer to Kolchinsky & Tracey (2017) for the proof. In addition to Bhattacharyaa distance, we applied PaiDEs with KL divergence as another proposed improvement to Monte Carlo estimation.

## 5   NORMALIZING FLOW ENSEMBLES

In this study, we utilize an ensemble technique named Nflows Base, which has previously shown robust performance in estimating both aleatoric and epistemic uncertainty on robotic datasets by leveraging normalizing flows (NFs) to create ensembles (Berry & Meger, 2023). PaiDEs can be employed with any ensemble possessing probabilistic outputs and closed-form distributional distance between ensemble components.

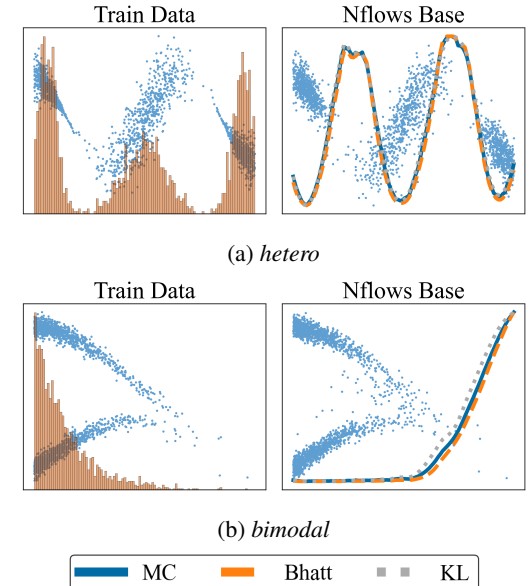

(a) *hetero*

(b) *bimodal*

### 5.1   NFLOWS BASE

NFs have been classically applied to unsupervised tasks (Tabak & Vanden-Eijnden, 2010; Tabak & Turner, 2013; Rezende & Mohamed, 2015), though NFs have been adapted to a supervised learning setting (Winkler et al., 2019; Ardizzone et al., 2019). Using the structure of Winkler et al. (2019) one can define a supervised NF as,

$$p_{y|x}(y|x) = p_{b|x}(g_\theta^{-1}(y, x))$$
$$\times |\det(J(g_\theta^{-1}(y, x)))|,$$
$$\log(p_{y|x}(y|x)) = \log(p_{b|x}(g_\theta^{-1}(y, x)))$$
$$+ \log(|\det(J(g_\theta^{-1}(y, x)))|),$$

where $p_{y|x}$ is the output distribution, $p_{b|b}$ is the base distribution, $J$ refers to the Jacobian, and $g_\theta^{-1} : y \times x \mapsto b$ is the bijective mapping. For a more comprehensive review of NFs, refer to

Figure 2: In the right graphs, the blue dots are sampled from Nflows Base and the 3 lines depict epistemic uncertainty corresponding to different estimators. The left graphs depicts the ground-truth data as the blue dots and its corresponding density as the orange histogram. Note the legend refers to the lines in the right graphs.

Papamakarios et al. (2021). Nflows Base creates an ensemble in the base distribution,

$$p_{y|x,\theta}(y|x, \theta) = f_{\theta_j}(y, x) = p_{b|x,\theta}(g_\theta^{-1}(y, x))|\det(J(g_\theta^{-1}(y, x)))|,$$

where $p_{b|x,\theta}(b|x, \theta) = N(\mu_{\theta,x}, \Sigma_{\theta,x})$, $\mu_{\theta,x}$ and $\Sigma_{\theta,x}$ denote the mean and covariance conditioned on both $x$ and $\theta$. These parameters are modeled using a neural network with fixed dropout masks to establish an ensemble and ensemble diversity is created by randomization and bootstrapping. By constructing the ensemble within the base distribution, we can leverage closed-form pairwise-distance formulae.

Berry & Meger (2023) showed that Nflows Base outperforms previous methods when estimating epistemic uncertainty, as the aleatoric uncertainty from Equation (2) can be estimated in the base distribution space and therefore allow for aleatoric uncertainty to be computed analytically. This does not apply to the other quantity of Equation (2), total uncertainty, and thus samples still need to be drawn in order to estimate epistemic uncertainty.

## 6 EPISTEMIC UNCERTAINTY ESTIMATION WITH PAIDES

### 6.1 ESTIMATORS

As mentioned in Section 3, the quantity of interest is mutual information rather than entropy. By applying our definition of PaiDEs to Equation (2), we obtain the following expression:

$$\hat{I}_\rho(y, \theta) = \hat{H}_\rho(y|x) - E_{p(w)}\left[H(y|x, \theta)\right] = -\sum_{i=1}^{M} \pi_i \ln \sum_{j=1}^{M} \pi_j \exp\left(-D(p_i \parallel p_j)\right), \quad (11)$$

as $E_{p(\theta)}\left[H(y|x, \theta)\right] = H(y|x, \theta)$. **PaiDEs provide a succinct estimator that can estimate epistemic uncertainty with only the pairwise distances between components, thus eliminating reliance on sample-based techniques.** We propose the following specific estimators:

$$\hat{I}_{Bhatt}(y, \theta) = -\sum_{i=1}^{M} \pi_i \ln \sum_{j=1}^{M} \pi_j \exp\left(-D_{Bhatt}(p_i \parallel p_j)\right),$$

$$\hat{I}_{KL}(y, \theta) = -\sum_{i=1}^{M} \pi_i \ln \sum_{j=1}^{M} \pi_j \exp\left(-D_{KL}(p_i \parallel p_j)\right),$$

where $D_{Bhatt}(p_i \parallel p_j)$ and $D_{KL}(p_i \parallel p_j)$ are defined for Gaussians in Appendix A.1. Note that our proposed estimators can be applied to any ensemble model whose output distributions have closed-form pairwise-distances as such, we have included experiments using probabilistic network ensembles (PNEs) in Appendix A.4.

### 6.2 COMBINATION OF PAIDES & NFLOWS BASE

Berry & Meger (2023) demonstrate that estimating Equation (2) in the base distribution is equivalent to estimating it in the output distribution. **Consequently, by combining Nflows Base and PaiDEs, we construct an expressive non-parametric model capable of capturing intricate aleatoric uncertainty in the output distribution while efficiently estimating epistemic uncertainty in the base distribution.** Unlike previously proposed methods, we are able to estimate epistemic uncertainty without taking a single sample. Figure 1 shows an example of the distributional pairs that need to be considered in order to estimate epistemic uncertainty for an Nflows Base model with 3 components.

## 7 EXPERIMENTAL RESULTS

To evaluate our method, we tested each PaiDE (KL $\hat{I}_{KL}(y, \theta)$ and Bhatt $\hat{I}_{Bhatt}(y, \theta)$) on two 1D environments, as has been previously proposed in the literature (Depeweg et al., 2018). Additionally, we present 4 multi-D environments. In contrast to previous papers (Berry & Meger, 2023), we increased the number of dimensions by more than an order of magnitude, from 11 to 257, to demonstrate the utility of PaiDEs in higher dimensions. The ensembles used in our experiments were constructed by randomly initializing the weights and creating bootstrapped samples of the training dataset. Also note that, for all experiments, the model components are assumed to be uniform, $\pi_j = \frac{1}{M}$, independent of $x$. In addition, all model hyper-parameters are contained in Appendix A.1 and the code can be found at (**added upon publication**).

### 7.1 DATA

We evaluated PaiDEs on two 1D benchmarks, *hetero* and *bimodal*. The ground-truth data for *hetero* and *bimodal* can be seen in Figure 2 on the left graphs with the blue dots with the orange bar chart corresponding to the density. For *hetero*, there are two regions with low density (2 and -2). In these regions, we would expect a model to have high epistemic uncertainty. For *bimodal*, the number of data points drops off as x increases, thus we would expect a model to have epistemic uncertainty grow as x does. All details for data generation are contained in Appendix A.2.

In addition to the 1D environments, we tested our methods over four multi-dimensional environments (*Pendulum-v0*, *Hopper-v2*, *Ant-v2*, and *Humanoid-v2*) (Todorov et al., 2012). Replay buffers were

Table 1: Mean RMSE on the test set for the last ($100^{th}$) Acquisition Batch for Nflows Base. Experiments were across ten different seeds and the results are expressed as mean plus minus one standard deviation with results that are statistically significant highlighted.

| Env. | Output Dim. | Random | MC | KL | Bhatt |
|---|---|---|---|---|---|
| *hetero* | 1 | $1.6 \pm 0.19$ | $1.58 \pm 0.32$ | $\mathbf{1.42} \pm 0.16$ | $1.43 \pm 0.18$ |
| *bimodal* | 1 | $6.4 \pm 0.62$ | $6.01 \pm 0.04$ | $6.01 \pm 0.04$ | $\mathbf{6.0} \pm 0.04$ |
| *Pendulum-v0* | 3 | $0.55 \pm 0.17$ | $\mathbf{0.09} \pm 0.02$ | $0.11 \pm 0.03$ | $0.12 \pm 0.04$ |
| *Hopper-v2* | 11 | $1.58 \pm 0.3$ | $0.61 \pm 0.05$ | $\mathbf{0.53} \pm 0.05$ | $0.56 \pm 0.05$ |
| *Ant-v2* | 32 | $2.16 \pm 0.06$ | $2.3 \pm 0.09$ | $\mathbf{2.06} \pm 0.08$ | $2.11 \pm 0.1$ |
| *Humanoid-v2* | 257 | $8.06 \pm 1.63$ | $7.78 \pm 1.41$ | $\mathbf{3.88} \pm 1.47$ | $4.96 \pm 2.76$ |

$p < 0.05$    $p < 0.01$    $p < 0.001$

gathered from an agent and the dynamics model for each environment was modeled, $f_\theta(s_t, a_t) = \hat{s}_{t+1}$. We evaluated on multi-dimensional environments because they are routinely used as benchmarks and provide us a higher dimensional output space to validate our methods. Also note that, for *Ant-v2* and *Humanoid-v2*, the dimensions representing their contact forces were eliminated as Mujoco-v2 had a bug, always returning zero for those dimensions[1].

## 7.2 1D EXPERIMENTS

Our 1D environments provide empirical proof that PaiDEs can accurately measure epistemic uncertainty. Figure 2 depicts that both KL and Bhatt are proficient at estimating the epistemic uncertainty as each method shows an increase in epistmeic uncertainty around 2 and -2 on the *hetero* setting. This can be seen from the orange and gray lines. KL and Bhatt perform indistinguishably from MC, as shown by the blue line.

A similar pattern can be seen for the *bimodal* setting in Figure 2, which shows that both Bhatt and KL can accurately capture epistemic uncertainty. Each estimator shows the pattern of increasing epistemic uncertainty where the data is more scarce. Both examples show accurate epistemic uncertainty estimation with no loss in aleatoric uncertainty representation, as demonstrated in the right graphs in Figure 2: the blue dots closely match the blue dots on their corresponding left graphs.

## 7.3 ACTIVE LEARNING

While the 1D experiments provide evidence of PaiDEs' effectiveness for estimating epistemic

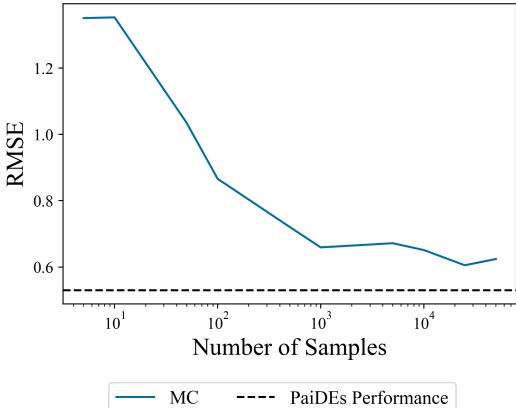

Figure 3: RMSE on the test set at the $100^{th}$ acquisition batch of the MC estimator on the *Hopper-v2* environment as the number of samples increases. Experiment run across 10 seeds and the mean is being reported.

uncertainty, the active learning experiments extend this evaluation to higher-dimensional data. Nflows Base started with 100 or 200 data points depending on the setting. At the end of each training epoch, the MC estimator sampled 1,000 unseen inputs and estimated their epistemic uncertainties, except for the *Humanoid-v2* environment where only 100 new inputs were sampled due to computational

---

[1]More information can be found here: https://github.com/openai/gym/issues/1541.

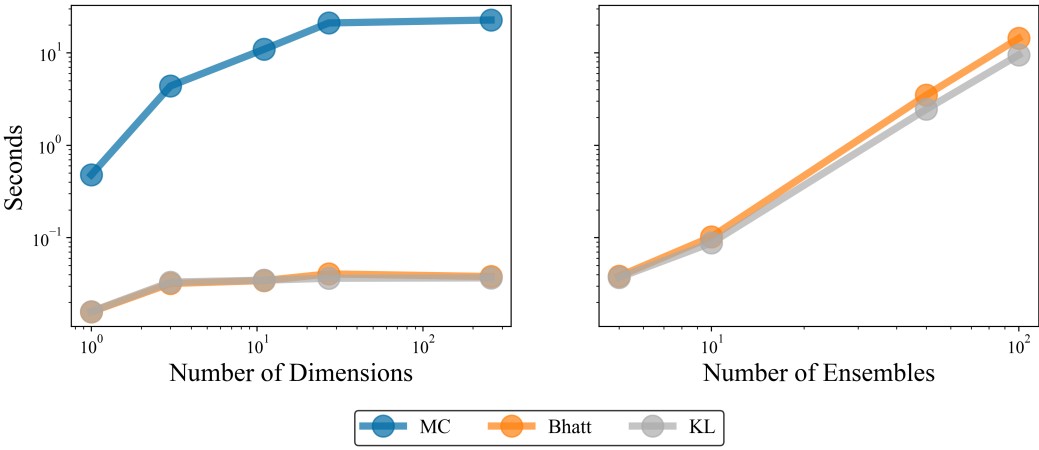

Figure 4: On the left, the amount of time taken for each estimator across the different settings (1, 3, 11, 27, 257 dimensions). On the right, the amount of time taken for PaiDEs as the number of ensemble components increases for the 257 dimensional setting. Results are averaged over 10 seeds and shown on a log scale.

constraints. On the other hand, PaiDEs sampled 10,000 new inputs and estimated their epistemic uncertainties for each environment. **This highlights one advantage PaiDEs over MC estimators, as PaiDEs are able to estimate epistemic uncertainty over larger regions at lower computational cost than their MC counterparts.** Upon estimating epistemic uncertainty, the 10 data points with the highest epistemic uncertainty (50 data points for *Humanoid-v2*) were added to the training set. Additionally, the root mean squared error (RMSE) on the test set was calculated at each acquisition batch.

Table 1 displays the performance of each estimator on the $100^{th}$ acquisition batch. In each environment, we conducted a Welch's t-test that compares both PaiDE estimators against the two baselines. Note that we included a Holm–Bonferroni correction to control the family-wise error rate (FWER), for more information refer to Appendix A.6. **For each data setting, the PaiDEs reach lower or comparable RMSEs to MC estimators, thus demonstrating that PaiDEs can be used to estimate epistemic uncertainty. In addition, PaiDEs are more effective in higher dimensions as can be seen by the fact that PaiDEs outperform MC estimates in statically significant manner for *Humanoid-v2* and *Ant-v2*.** A random acquisition function was included as a baseline.

To conduct a more in-depth analysis of our proposed method, we compared PaiDEs to MC estimators with a varying sample size in the *Hopper-v2* environment. We expected that MC estimators would perform on par with PaiDEs with a sufficient number of samples. However, as illustrated in Figure 3, MC estimators fell short of achieving the same level of performance as PaiDEs in this particular scenario. This suggests that, taking into consideration hardware constraints, PaiDEs begin to outperform MC estimators when dealing with 11 dimensional outputs.

## 7.4 TIME ANALYSIS & LIMITATIONS

In addition to benchmarking PaiDEs on active learning experiments, we provide an analysis of the time gains across our experiments. The left hand side of Figure 4 depicts the speed increase that can be gained using PaiDEs over an MC approach. A 1-2 order magnitude of improvement can be seen. The estimates are obtained from the active learning experiments, and the number of dimensions corresponds to each of the environments.

A weakness of PaiDEs is that as the number of components increases, the computational cost rises. Therefore algorithms like MC dropout (Gal & Ghahramani, 2016) may not be suitable. In the instance where the distance is not symmetric, KL-divergence, $M^2 - M$ pairwise-distances need to be computed. For symmetric distances, such as Bhattacharyaa distance, only $\frac{M^2 - M}{2}$ distances need

to computed. The right hand side of Figure 4, shows an analysis of the time taken as the number of ensembles grow. Note that for Bhatt, the time costs could be improved upon using the symmetry logic described as the results shown calculated all pairwise-distances. Despite the growing complexity of PaiDEs with the number of components, this is normally not a problem for deep learning ensembles as they have a relatively low number (5-10) of components (Osband et al., 2016; Chua et al., 2018).

An additional limitation is the bias introduced by PaiDEs, which MC estimators do not suffer from. It is essential to note that, in the context of active learning, epistemic uncertainty serves as a relative quantity for comparing potential acquisition points. The introduction of bias from PaiDEs does not impact the relative relationship of epistemic uncertainty between different data points. We demonstrate that the relative relationship of epistemic uncertainty remains intact in Appendix A.3.

## 8 RELATED WORK

Researchers have employed Bayesian neural networks alongside information-based criteria for active learning in image classification problems (Gal et al., 2017; Kendall & Gal, 2017; Kirsch et al., 2019). These studies utilize epistemic uncertainty estimation with MC dropout to gauge uncertainty in image classification tasks. In contrast, our research focuses on estimating uncertainty within a continuous output space. Our experiments encompassed tasks where the output spans continuous distributions for 1 to 257 dimensions, as opposed to the aforementioned methods that primarily address classification problems with a 1D categorical output.

In addition to Bayesian methods, ensembles have been harnessed for epistemic uncertainty estimation (Lakshminarayanan et al., 2017; Choi et al., 2018; Chua et al., 2018). Specifically related to our work, ensembles have been leveraged to quantify epistemic uncertainty in regression problems and active learning (Depeweg et al., 2018; Postels et al., 2020; Berry & Meger, 2023). Depeweg et al. (2018) employed Bayesian neural networks to model mixtures of Gaussians and demonstrated their ability to measure uncertainty in low-dimensional environments (1-2D). Building upon this foundation, Postels et al. (2020) and Berry & Meger (2023) extended the research by developing efficient Normalizing Flow (NF) ensemble models that effectively captured epistemic uncertainty. Our work advances this line of research by eliminating the need for sampling to estimate epistemic uncertainty, resulting in a faster and more effective method, especially in higher dimensions.

Entropy estimators, which do not rely on sampling, is an active area of research (Jebara & Kondor, 2003; Jebara et al., 2004; Huber et al., 2008; Kolchinsky & Tracey, 2017). Kulak et al. (2021) and Kulak & Calinon (2021) demonstrated the utility of Pairwise-Distance Estimators (PaiDEs) within Bayesian contexts, employing PaiDEs to estimate conditional predictive posterior entropy. In contrast, our approach provides a more general estimate of epistemic uncertainty, as defined in Equation (2), which can be applied to both ensemble and Bayesian methods. Furthermore, our method is adaptable to flexible deep learning models, a capability that was previously unavailable in the approach presented by Kulak et al. (2021) and Kulak & Calinon (2021).

Several methods have emerged in the literature for estimating epistemic uncertainty without relying on sampling techniques (Van Amersfoort et al., 2020; Charpentier et al., 2020). Both Van Amersfoort et al. (2020) and Charpentier et al. (2020) focus on classification tasks with 1D categorical outputs. Charpentier et al. (2021) extends the work of Charpentier et al. (2020) to regression tasks but is limited to modeling outputs as members of the exponential family. In contrast, our approach can handle more complex output distributions by directly considering the outputs from Normalizing Flows (NFs). This flexibility is particularly valuable in scenarios involving intricate non-linear robotic dynamics, as demonstrated in our experiments.

## 9 CONCLUSIONS

In this study, we introduced two epistemic uncertainty estimators and applied them to expressive ensemble models. We depicted how our method can be used to more efficiently quantify uncertainty by leveraging closed-form pairwise-distance instead of sampling. This led to improvements in computational speed and accuracy, especially in larger dimensions. We addressed the problem of epistemic uncertainty estimation in high-dimensional problems by building effective epistemic uncertainty estimators without sampling.

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

## A APPENDIX

### A.1 COMPUTE AND HYPER-PARAMETER DETAILS

The Nflows Base model employed one nonlinear transformation, $g$, with a single hidden layer containing 20 units, utilizing cubic spline flows as per Durkan et al. (2019). The base network consisted of two hidden layers, each comprising 40 units with ReLU activation functions. It is important to note that all base distributions were Gaussian. The PNEs adopted an architecture of three hidden layers each with 50 units and ReLU activation functions. Model hyperparameters remained consistent across all experiments. Training was conducted using 16GB RAM on Intel Gold 6148 Skylake @ 2.4 GHz CPUs and NVidia V100SXM2 (16G memory) GPUs. For each experimental setting, PNEs and Nflows

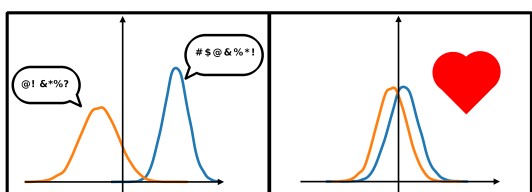

Figure 5: The amount of epistemic uncertainty in an ensemble of probabilistic learners is high when the mixture components disagree (left) and low when there is agreement (right).

Base were executed with five ensemble components. The MC estimator sampled 1000 and 5000 points for Nflows Base and PNEs, respectively, for each $x$ conditioned on. The nflows library (Durkan et al., 2020) was employed with minor modifications.

Note that for the Bhatt estimator, the Bhattacharyya distance between two Gaussians is,

$$D_B(p_i||p_j) = \frac{1}{8}(\mu_{i|x} - \mu_{j|x})^\mathrm{T}\Sigma^{-1}(\mu_{i|x} - \mu_{j|x}) + \frac{1}{2}\ln\left(\frac{\det\Sigma}{\sqrt{\det\Sigma_{i|x}\det\Sigma_{j|x}}}\right),$$

$$\Sigma = \frac{\Sigma_{i|x} + \Sigma_{j|x}}{2}.$$

Also note that for the KL estimator, the KL divergence between two Gaussians is,

$$D_{KL}(p_i \parallel p_j) = \frac{1}{2}\left(\mathrm{tr}(\Sigma_{j|x}^{-1}\Sigma_{i|x}) - D + \ln\left(\frac{\det\Sigma_{j|x}}{\det\Sigma_{i|x}}\right) + (\mu_{j|x} - \mu_{i|x})^\mathrm{T}\Sigma_{j|x}^{-1}(\mu_{j|x} - \mu_{i|x})\right),$$

where $\mathrm{tr}(\cdot)$ refers to the trace of a matrix. Figure 5 illustrates an example where a pair of committee members agree (i.e., small distributional distance) on the right, and another pair disagree (i.e., large distributional distance) on the left.

### A.2 DATA

The *hetero* dataset was generated using a two step process. Firstly, a categorical distribution with three values was sampled, where $p_i = \frac{1}{3}$. Secondly, $x$ was drawn from one of three different Gaussian distributions ($N(-4, \frac{2}{5})$, $N(0, \frac{9}{10})$, $N(4, \frac{2}{5})$) based on the value of the categorical distribution. The corresponding $y$ was then generated as follows:

$$y = 7\sin(x) + 3z\left|\cos\left(\frac{x}{2}\right)\right|.$$

On the other hand, the *bimodal* dataset was created by sampling $x$ from an exponential distribution with parameter $\lambda = 2$, and then sampling $n$ from a Bernoulli distribution with $p = 0.5$. Based on the value of $n$, the $y$ value was determined as:

$$y = \begin{cases} 10\sin(x) + z & n = 0 \\ 10\cos(x) + z + 20 - x & n = 1 \end{cases}.$$

Note that for both *bimodal* and *hetero* data $z \sim N(0, 1)$.

Regarding the multi-dimensional environments, namely *Pendulum-v0*, *Hopper-v2*, *Ant-v2*, and *Humanoid-v2*, the training sets and test sets were collected using different approaches. The training sets were obtained by applying a random policy, while the test sets were generated using an expert policy. This methodology was employed to ensure diversity between the training and test datasets. Notably, the OpenAI Gym library was utilized, with minor modifications (Brockman et al., 2016).

## A.3 ADDITIONAL RESULTS

We provide a analysis of the bias introduced by PaiDEs in Figure 6. Note that in Figure 2 the epistemic uncertainty values mapped to 0-1 via a min-max normalization. The right two plots show that PaiDEs introduce some bias for both 1D settings, though this is not a problem in an active learning setting as we only care about a point's uncertainty relative to other points and this relationship is preserved. In addition to Table 1, we provide Figure 7 to show the entire active leaning curve and Table 2 detailing more acquisition batches. The pattern of PaiDEs outperforming or perform similarly to baselines holds across all environments and acquisition batches. An additional metric we evaluated on was log-likelihood, as the log-likelihood has been shown to be a proper scoring rule (Harakeh & Waslander, 2021). The results are shown in Figure 8, with 10 seeds being run and mean standard deviation being reported. As can be seen, the KL and Bhatt estimators perform similarly or better than the MC estimator on all environments. Note that due to the high-dimensional setting of *Humanoid-v2* the log-likelihood for each estimator did not improve as data was added. This is due to the fact, that in order to calculate the log-likelihood of the ensemble one needs to cal-

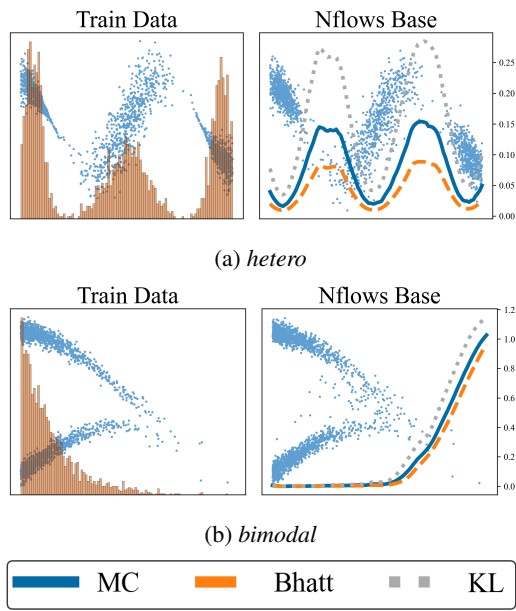

(a) *hetero*

(b) *bimodal*

Figure 6: The same plots as Figure 2 except there is no mapping applied and the corresponding estimates for $I(y, W)$ are shown the right y-axes for the right plots.

culate the likelihood of each individual model and then sum them together. In this process we are more likely to run into an issue of values being rounded to zero as we cannot store enough digits. As in Table 1, we have recorded certain acquisition batches in Table 3. Lastly, we provide a comparison on *Hopper-v2* of the MC estimator to PaiDEs as the number samples drawn is increased in Figure 9.

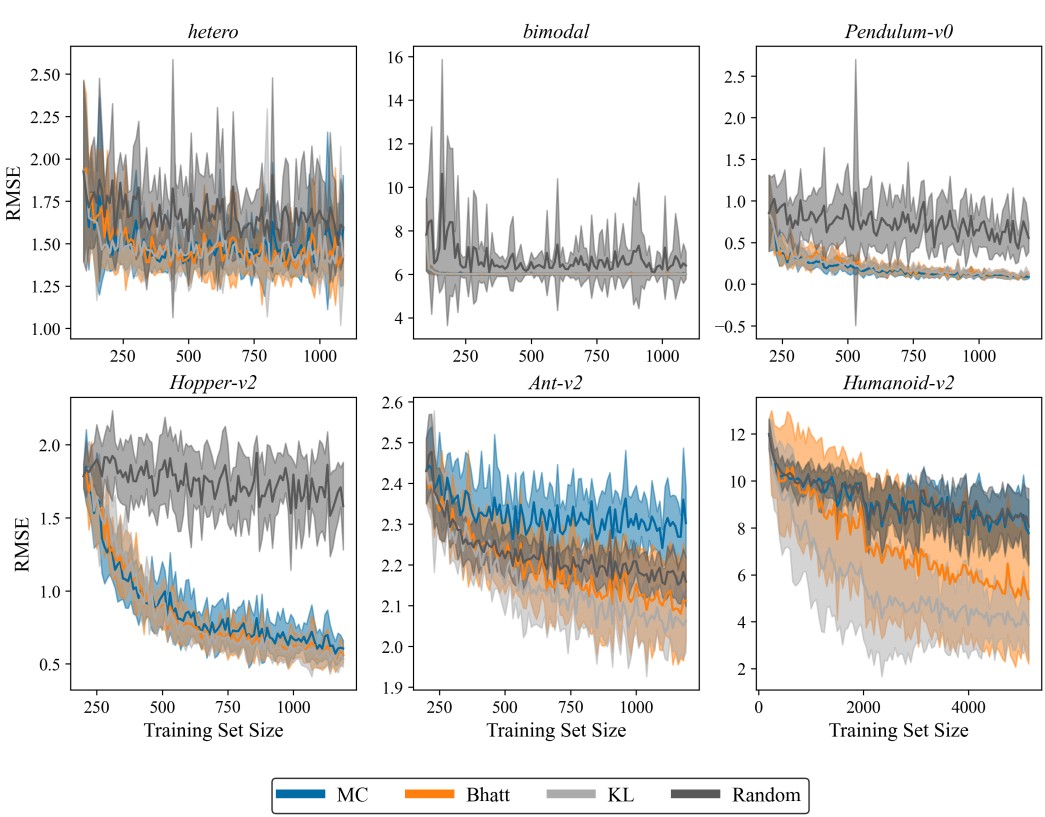

Figure 7: Mean RMSE on the test set as data was added to the training set for Nflows Base.

Table 2: Mean RMSE on the test set for certain Acquisition Batches for Nflows Base. Experiments were across ten different seeds and the results are expressed as mean plus minus one standard deviation.

| Env | Acq. Batch | Random | MC | KL | Bhatt |
|---|---|---|---|---|---|
| *hetero* | 10 | $1.78 \pm 0.21$ | $1.54 \pm 0.14$ | $\mathbf{1.49} \pm 0.15$ | $1.7 \pm 0.35$ |
| | 25 | $1.65 \pm 0.42$ | $1.41 \pm 0.11$ | $1.46 \pm 0.1$ | $\mathbf{1.39} \pm 0.11$ |
| | 50 | $1.67 \pm 0.24$ | $1.4 \pm 0.12$ | $\mathbf{1.45} \pm 0.12$ | $1.48 \pm 0.25$ |
| | 100 | $1.6 \pm 0.19$ | $1.58 \pm 0.32$ | $\mathbf{1.42} \pm 0.16$ | $1.43 \pm 0.18$ |
| *bimodal* | 10 | $8.39 \pm 3.38$ | $6.02 \pm 0.04$ | $\mathbf{6.01} \pm 0.05$ | $\mathbf{6.01} \pm 0.05$ |
| | 25 | $6.1 \pm 0.08$ | $6.02 \pm 0.05$ | $6.02 \pm 0.05$ | $\mathbf{6.01} \pm 0.04$ |
| | 50 | $6.57 \pm 0.72$ | $\mathbf{6.01} \pm 0.04$ | $\mathbf{6.01} \pm 0.04$ | $\mathbf{6.01} \pm 0.04$ |
| | 100 | $6.4 \pm 0.62$ | $6.01 \pm 0.04$ | $6.01 \pm 0.04$ | $\mathbf{6.0} \pm 0.04$ |
| *Pendulum-v0* | 10 | $0.67 \pm 0.28$ | $0.33 \pm 0.1$ | $\mathbf{0.32} \pm 0.08$ | $0.37 \pm 0.19$ |
| | 25 | $0.73 \pm 0.24$ | $\mathbf{0.22} \pm 0.07$ | $0.3 \pm 0.2$ | $0.3 \pm 0.16$ |
| | 50 | $0.65 \pm 0.42$ | $\mathbf{0.16} \pm 0.05$ | $0.17 \pm 0.05$ | $0.18 \pm 0.06$ |
| | 100 | $0.55 \pm 0.17$ | $\mathbf{0.09} \pm 0.02$ | $0.11 \pm 0.03$ | $0.12 \pm 0.04$ |
| *Hopper-v2* | 10 | $1.91 \pm 0.22$ | $\mathbf{1.26} \pm 0.18$ | $1.58 \pm 0.21$ | $1.54 \pm 0.23$ |
| | 25 | $1.89 \pm 0.19$ | $\mathbf{0.92} \pm 0.09$ | $1.1 \pm 0.14$ | $0.98 \pm 0.11$ |
| | 50 | $1.75 \pm 0.17$ | $0.78 \pm 0.12$ | $0.71 \pm 0.08$ | $\mathbf{0.69} \pm 0.06$ |
| | 100 | $1.58 \pm 0.3$ | $0.61 \pm 0.05$ | $\mathbf{0.53} \pm 0.05$ | $0.56 \pm 0.05$ |
| *Ant-v2* | 10 | $2.29 \pm 0.05$ | $2.38 \pm 0.09$ | $\mathbf{2.26} \pm 0.06$ | $2.28 \pm 0.07$ |
| | 25 | $2.25 \pm 0.07$ | $2.34 \pm 0.1$ | $2.26 \pm 0.1$ | $\mathbf{2.22} \pm 0.09$ |
| | 50 | $2.17 \pm 0.04$ | $2.28 \pm 0.07$ | $\mathbf{2.11} \pm 0.07$ | $2.14 \pm 0.1$ |
| | 100 | $2.16 \pm 0.06$ | $2.3 \pm 0.09$ | $\boxed{\mathbf{2.06} \pm 0.08}$ | $2.11 \pm 0.1$ |
| *Humanoid-v2* | 10 | $10.22 \pm 0.64$ | $10.11 \pm 1.05$ | $\mathbf{8.54} \pm 2.82$ | $10.58 \pm 1.8$ |
| | 25 | $9.78 \pm 0.51$ | $10.17 \pm 0.57$ | $\boxed{\mathbf{6.61} \pm 2.39}$ | $8.54 \pm 2.89$ |
| | 50 | $8.74 \pm 1.03$ | $8.75 \pm 1.15$ | $\boxed{\mathbf{4.87} \pm 2.36}$ | $7.08 \pm 3.22$ |
| | 100 | $8.06 \pm 1.63$ | $7.78 \pm 1.41$ | $\boxed{\mathbf{3.88} \pm 1.47}$ | $\boxed{4.96 \pm 2.76}$ |

$p < 0.05$    $p < 0.01$    $p < 0.001$

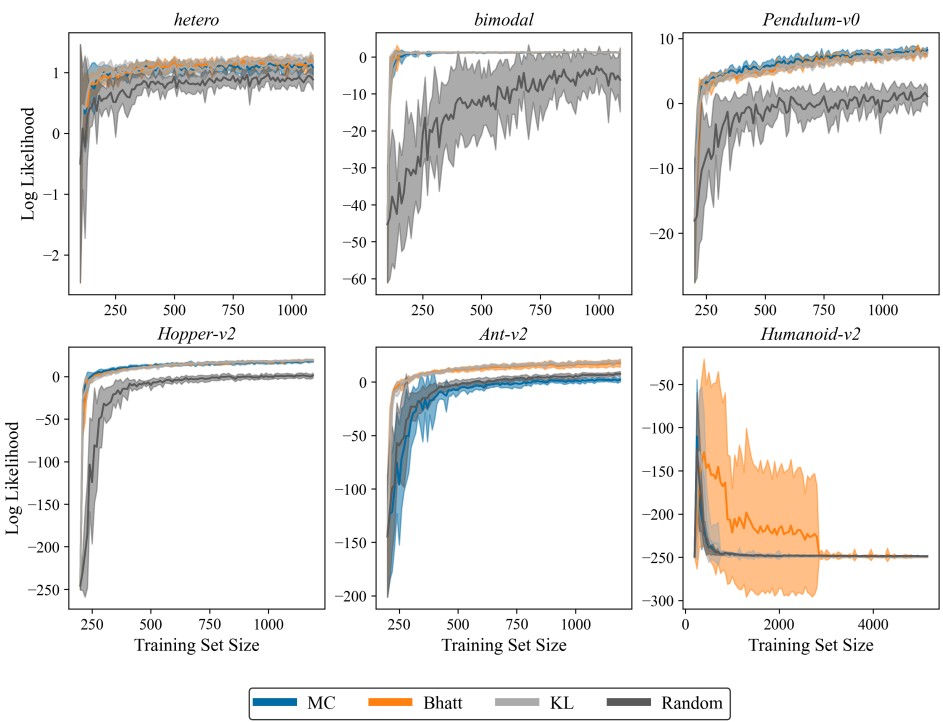

Figure 8: Mean Log Likelihood on test set as data was added to the training sets for Nflows Base.

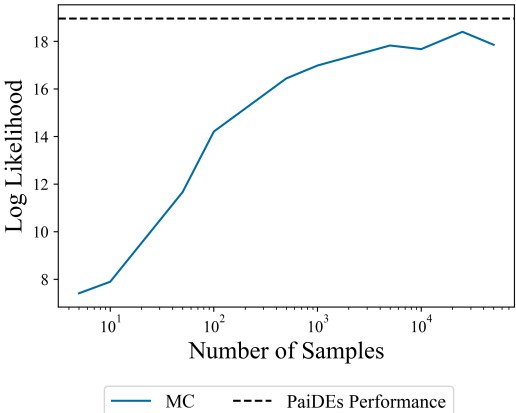

Figure 9: Log-Likelihood on the test set at the $100^{th}$ acquisition batch of the MC estimator on the *Hopper-v2* environment as the number of samples increases for Nflows Base. Experiment run across 10 seeds and the mean is being reported.

Table 3: Log Likelihood of a held out test set during training at different acquisition batches for Nflows Base. Experiments were across ten different seeds and the results are expressed as mean plus minus one standard deviation.

| Env | Acq. Batch | Random | MC | KL | Bhatt |
|---|---|---|---|---|---|
| *hetero* | 10 | $0.53 \pm 0.26$ | $0.95 \pm 0.14$ | $\textbf{0.98} \pm 0.19$ | $0.9 \pm 0.14$ |
| | 25 | $0.79 \pm 0.18$ | $\textbf{1.08} \pm 0.14$ | $1.05 \pm 0.15$ | $1.07 \pm 0.11$ |
| | 50 | $0.8 \pm 0.15$ | $\textbf{1.1} \pm 0.04$ | $1.07 \pm 0.16$ | $\textbf{1.1} \pm 0.12$ |
| | 100 | $0.88 \pm 0.17$ | $1.08 \pm 0.12$ | $1.17 \pm 0.1$ | $\textbf{1.18} \pm 0.11$ |
| *bimodal* | 10 | $-30.57 \pm 12.61$ | $0.86 \pm 0.51$ | $\textbf{1.11} \pm 0.1$ | $\textbf{1.11} \pm 0.1$ |
| | 25 | $-16.55 \pm 9.04$ | $1.22 \pm 0.08$ | $1.17 \pm 0.1$ | $\textbf{1.27} \pm 0.09$ |
| | 50 | $-10.85 \pm 8.5$ | $1.18 \pm 0.13$ | $1.2 \pm 0.1$ | $\textbf{1.21} \pm 0.11$ |
| | 100 | $-6.19 \pm 8.66$ | $\textbf{1.27} \pm 0.14$ | $1.26 \pm 0.1$ | $1.26 \pm 0.14$ |
| *Pendulum* | 10 | $-3.23 \pm 4.78$ | $3.67 \pm 1.01$ | $3.52 \pm 0.9$ | $\textbf{3.8} \pm 0.86$ |
| | 25 | $-0.92 \pm 3.25$ | $\textbf{5.26} \pm 0.71$ | $4.99 \pm 0.86$ | $4.92 \pm 0.78$ |
| | 50 | $0.05 \pm 2.93$ | $\textbf{6.55} \pm 0.61$ | $6.3 \pm 0.64$ | $6.43 \pm 0.74$ |
| | 100 | $1.08 \pm 1.49$ | $\textbf{8.21} \pm 0.47$ | $7.79 \pm 0.55$ | $7.79 \pm 0.49$ |
| *Hopper* | 10 | $-47.1 \pm 36.42$ | $\textbf{4.53} \pm 4.25$ | $0.93 \pm 2.02$ | $2.76 \pm 3.22$ |
| | 25 | $-11.81 \pm 4.48$ | $\textbf{11.7} \pm 1.94$ | $9.34 \pm 1.34$ | $10.19 \pm 1.84$ |
| | 50 | $-3.33 \pm 3.01$ | $13.37 \pm 1.74$ | $\textbf{13.98} \pm 1.03$ | $13.89 \pm 0.9$ |
| | 100 | $1.15 \pm 2.84$ | $18.4 \pm 1.69$ | $18.86 \pm 1.45$ | $\textbf{18.96} \pm 0.99$ |
| *Ant* | 10 | $-32.85 \pm 22.01$ | $-50.75 \pm 34.25$ | $1.55 \pm 3.19$ | $\textbf{2.54} \pm 3.5$ |
| | 25 | $-4.95 \pm 4.69$ | $-11.23 \pm 5.67$ | $9.78 \pm 1.81$ | $\textbf{9.96} \pm 1.51$ |
| | 50 | $2.37 \pm 3.47$ | $-1.33 \pm 4.06$ | $\textbf{14.9} \pm 1.65$ | $14.64 \pm 2.71$ |
| | 100 | $7.52 \pm 2.39$ | $2.18 \pm 2.02$ | $\textbf{18.61} \pm 2.21$ | $17.67 \pm 3.85$ |
| *Humanoid* | 10 | $-244.8 \pm 3$ | $-244 \pm 1.3$ | $-234.1 \pm 21.6$ | $\textbf{-159.2} \pm 111$ |
| | 25 | $-247.8 \pm 0.9$ | $-248 \pm 0.9$ | $-247.6 \pm 3.5$ | $\textbf{-210.1} \pm 76.8$ |
| | 50 | $-248.6 \pm 0.6$ | $-248.3 \pm 0.9$ | $-249.3 \pm 0.5$ | $\textbf{-226.6} \pm 66.2$ |
| | 100 | $-248.8 \pm 0.4$ | $-248.8 \pm 0.4$ | $-248.7 \pm 0.2$ | $\textbf{-247.6} \pm 0.8$ |

■ $p < 0.05$ ■ $p < 0.01$ ■ $p < 0.001$

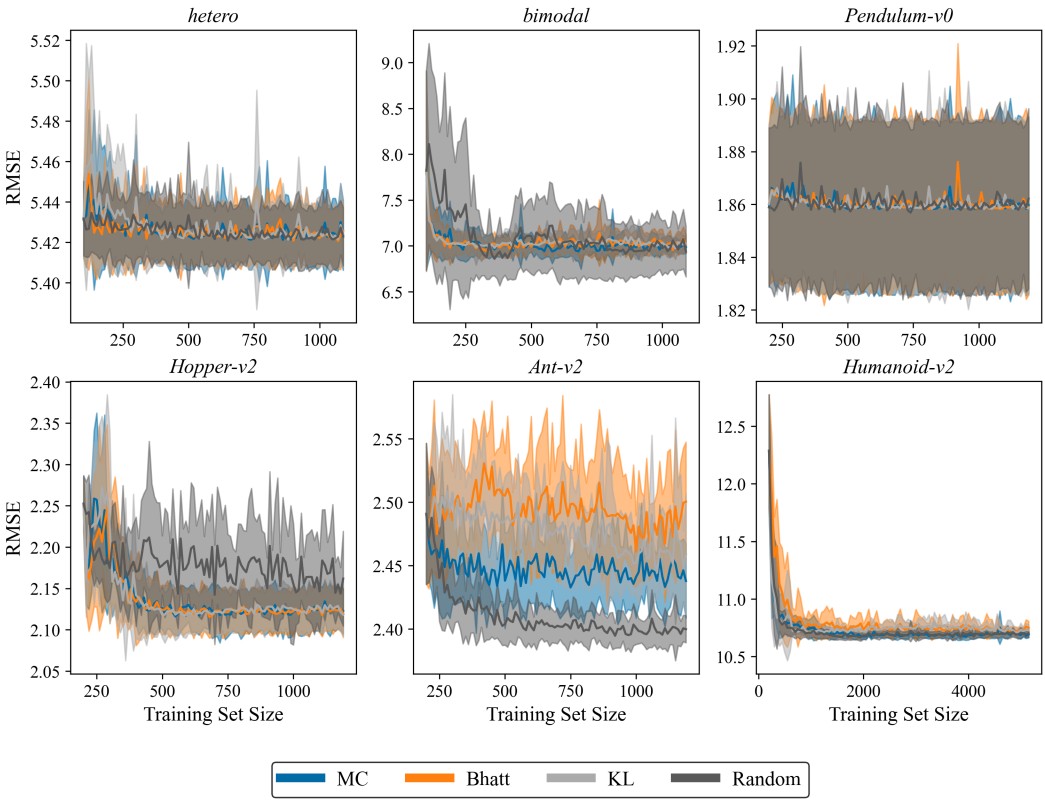

Figure 10: Mean RMSE on test set as data was added to the training sets for PNEs.

## A.4 PROBABILISTIC NETWORK ENSEMBLES

In addition to ensembles based on normalizing flows, we also explored the use of probabilistic network ensembles (PNEs) as a common approach for capturing uncertainty (Chua et al., 2018; Kurutach et al., 2018). The PNEs were constructed similarly to Nflows Base, employing fixed dropout masks, where each ensemble component modeled a Gaussian distribution. The models were trained using negative log likelihood, with weights randomly initialized and bootstrapped samples from the training set. Our findings paralleled those of Nflows Base, with the exception of RMSE improvement. We observed that PNEs lacked expressiveness as they only produced Gaussian outputs, limiting their ability to utilize epistemic uncertainty to enhance RMSE. These results are presented in Figures 10 and 13, as well as Tables 4 and 5. Furthermore, we have included the 1D graphs illustrating the performance of PNEs in *hetero* and *bimodal* in Figure 11. Similarly to before, we also provide a comparison of the MC estimator as the number of samples increased in Figure 12.

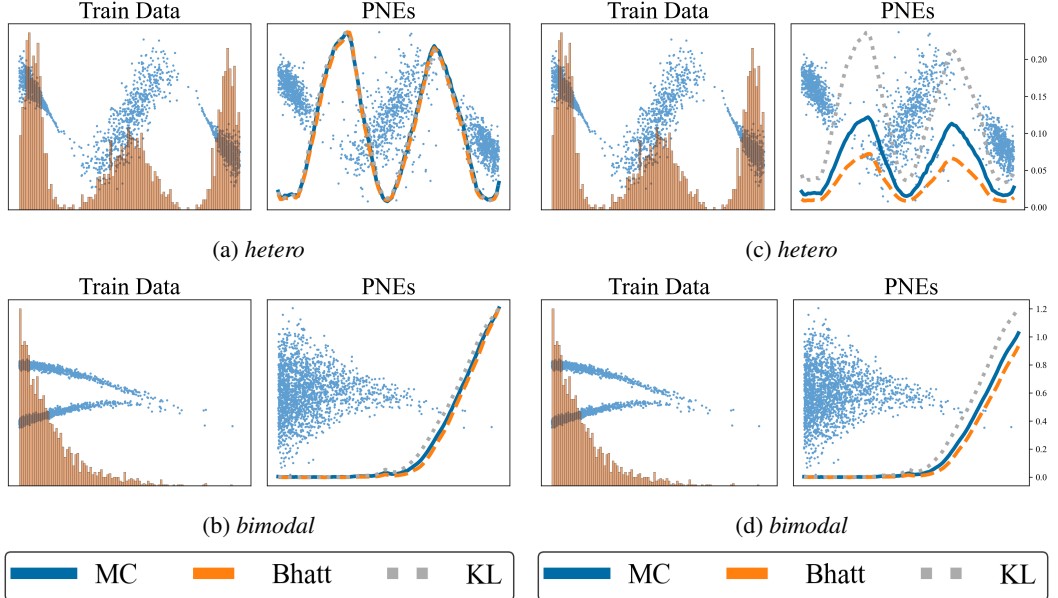

Figure 11: In the right graphs, the blue dots are sampled from PNEs and the 3 lines depict epistemic uncertainty corresponding to different estimators. The left graphs depicts the ground-truth data as the blue dots and its corresponding density as the orange histogram. Note the legend refers to the lines in the right graphs.

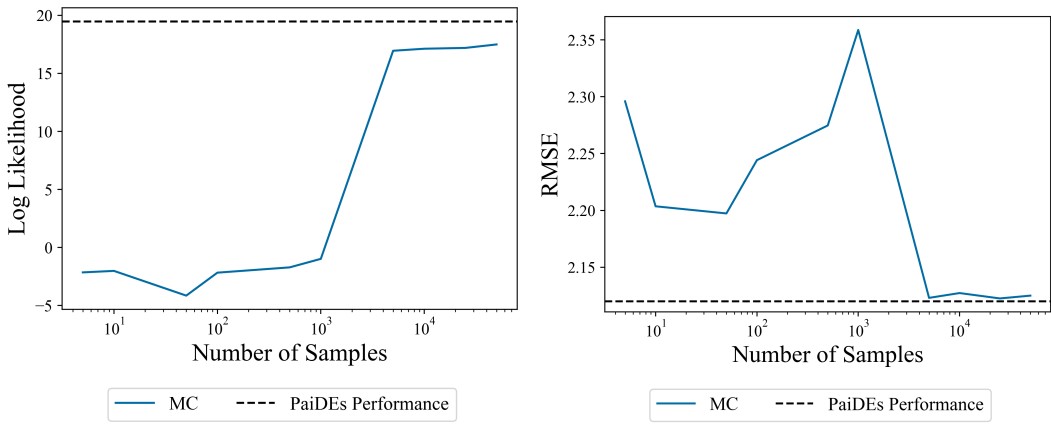

Figure 12: RMSE and Log-Likelihood on the test set at the $100^{th}$ acquisition batch of the MC estimator on the *Hopper-v2* environment as the number of samples increases for PNEs. Experiment run across 10 seeds and the mean is being reported.

Table 4: Mean RMSE on the test set for certain Acquisition Batches for PNEs. Experiments were across ten different seeds and the results are expressed as mean plus minus one standard deviation.

| Env | Acq. Batch | Random | MC | KL | Bhatt |
|---|---|---|---|---|---|
| *hetero* | 10 | $5.43 \pm 0.02$ | $5.44 \pm 0.02$ | $5.44 \pm 0.03$ | $5.43 \pm 0.02$ |
| | 25 | $5.43 \pm 0.02$ | $5.43 \pm 0.02$ | $5.43 \pm 0.03$ | $5.42 \pm 0.01$ |
| | 50 | $5.43 \pm 0.02$ | $5.42 \pm 0.01$ | $5.43 \pm 0.02$ | $5.42 \pm 0.01$ |
| | 100 | $5.42 \pm 0.01$ | $5.42 \pm 0.02$ | $5.42 \pm 0.02$ | $5.43 \pm 0.01$ |
| *bimodal* | 10 | $7.41 \pm 1.11$ | $7.18 \pm 0.3$ | $7.02 \pm 0.17$ | $6.98 \pm 0.19$ |
| | 25 | $7.02 \pm 0.38$ | $6.99 \pm 0.13$ | $6.98 \pm 0.1$ | $7.0 \pm 0.11$ |
| | 50 | $7.1 \pm 0.4$ | $6.95 \pm 0.08$ | $7.05 \pm 0.12$ | $7.04 \pm 0.1$ |
| | 100 | $6.94 \pm 0.28$ | $6.99 \pm 0.07$ | $7.02 \pm 0.08$ | $7.02 \pm 0.1$ |
| *Pendulum-v0* | 10 | $1.86 \pm 0.03$ | $1.87 \pm 0.04$ | $1.86 \pm 0.03$ | $1.86 \pm 0.03$ |
| | 25 | $1.86 \pm 0.03$ | $1.86 \pm 0.03$ | $1.86 \pm 0.03$ | $1.86 \pm 0.03$ |
| | 50 | $1.86 \pm 0.03$ | $1.86 \pm 0.03$ | $1.86 \pm 0.03$ | $1.86 \pm 0.03$ |
| | 100 | $1.86 \pm 0.03$ | $1.86 \pm 0.03$ | $1.86 \pm 0.03$ | $1.86 \pm 0.03$ |
| *Hopper-v2* | 10 | $2.19 \pm 0.05$ | $2.18 \pm 0.06$ | $2.27 \pm 0.12$ | $2.25 \pm 0.1$ |
| | 25 | $2.21 \pm 0.06$ | $2.12 \pm 0.03$ | $2.13 \pm 0.05$ | $2.13 \pm 0.04$ |
| | 50 | $2.2 \pm 0.06$ | $2.12 \pm 0.03$ | $2.12 \pm 0.03$ | $2.12 \pm 0.03$ |
| | 100 | $2.16 \pm 0.06$ | $2.12 \pm 0.03$ | $2.12 \pm 0.03$ | $2.12 \pm 0.03$ |
| *Ant-v2* | 10 | $2.42 \pm 0.02$ | $2.45 \pm 0.04$ | $2.49 \pm 0.04$ | $2.5 \pm 0.04$ |
| | 25 | $2.41 \pm 0.02$ | $2.43 \pm 0.02$ | $2.48 \pm 0.04$ | $2.51 \pm 0.05$ |
| | 50 | $2.4 \pm 0.01$ | $2.45 \pm 0.03$ | $2.48 \pm 0.03$ | $2.49 \pm 0.04$ |
| | 100 | $2.4 \pm 0.01$ | $2.44 \pm 0.03$ | $2.46 \pm 0.05$ | $2.5 \pm 0.05$ |
| *Humanoid-v2* | 10 | $10.74 \pm 0.07$ | $10.77 \pm 0.08$ | $10.72 \pm 0.01$ | $10.82 \pm 0.1$ |
| | 25 | $10.68 \pm 0.03$ | $10.69 \pm 0.05$ | $10.74 \pm 0.07$ | $10.79 \pm 0.13$ |
| | 50 | $10.71 \pm 0.04$ | $10.69 \pm 0.05$ | $10.77 \pm 0.09$ | $10.76 \pm 0.09$ |
| | 100 | $10.7 \pm 0.04$ | $10.69 \pm 0.03$ | $10.73 \pm 0.05$ | $10.75 \pm 0.06$ |

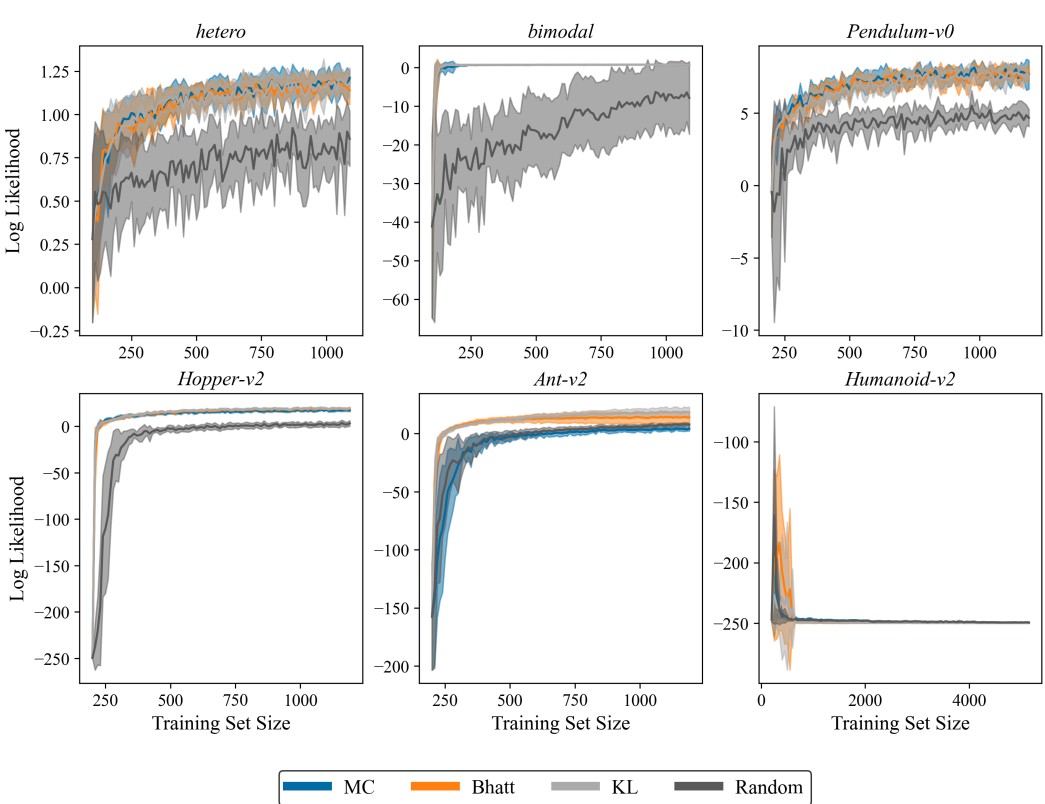

Figure 13: Mean Log Likelihood on the test set as data was added to the training sets for PNEs.

Table 5: Log Likelihood on the test set during training at different acquisition batches for PNEs. Experiments were across ten different seeds and the results are expressed as mean plus minus one standard deviation.

| Env | Acq. Batch | Random | MC | KL | Bhatt |
|---|---|---|---|---|---|
| *hetero* | 10 | $0.6 \pm 0.19$ | $\mathbf{0.89} \pm 0.18$ | $0.87 \pm 0.22$ | $0.88 \pm 0.16$ |
| | 25 | $0.56 \pm 0.26$ | $0.98 \pm 0.06$ | $1.01 \pm 0.08$ | $\mathbf{1.02} \pm 0.12$ |
| | 50 | $0.79 \pm 0.18$ | $\mathbf{1.15} \pm 0.09$ | $1.08 \pm 0.11$ | $1.11 \pm 0.1$ |
| | 100 | $0.86 \pm 0.16$ | $\mathbf{1.21} \pm 0.04$ | $1.18 \pm 0.09$ | $1.14 \pm 0.08$ |
| *bimodal* | 10 | $-27.37 \pm 16.56$ | $0.43 \pm 0.47$ | $\mathbf{0.63} \pm 0.06$ | $0.62 \pm 0.05$ |
| | 25 | $-21.15 \pm 8.54$ | $0.65 \pm 0.05$ | $\mathbf{0.67} \pm 0.03$ | $\mathbf{0.67} \pm 0.03$ |
| | 50 | $-16.08 \pm 9.12$ | $0.67 \pm 0.03$ | $0.69 \pm 0.02$ | $\mathbf{0.7} \pm 0.02$ |
| | 100 | $-7.91 \pm 9.41$ | $0.69 \pm 0.03$ | $\mathbf{0.7} \pm 0.03$ | $\mathbf{0.7} \pm 0.02$ |
| *Pendulum* | 10 | $3.43 \pm 0.9$ | $5.37 \pm 0.58$ | $4.86 \pm 0.34$ | $\mathbf{5.38} \pm 0.95$ |
| | 25 | $4.16 \pm 1.07$ | $\mathbf{6.55} \pm 1.05$ | $6.26 \pm 0.73$ | $6.44 \pm 0.77$ |
| | 50 | $4.34 \pm 1.2$ | $7.11 \pm 0.67$ | $\mathbf{7.52} \pm 0.41$ | $7.2 \pm 0.6$ |
| | 100 | $4.66 \pm 0.59$ | $7.65 \pm 0.99$ | $7.6 \pm 0.23$ | $\mathbf{7.78} \pm 0.53$ |
| *Hopper* | 10 | $-30.75 \pm 27.71$ | $\mathbf{9.19} \pm 1.65$ | $8.15 \pm 0.98$ | $7.98 \pm 1.43$ |
| | 25 | $-4.65 \pm 1.94$ | $13.21 \pm 1.83$ | $\mathbf{14.7} \pm 2.23$ | $13.83 \pm 1.9$ |
| | 50 | $-2.36 \pm 3.02$ | $15.91 \pm 1.38$ | $\mathbf{18.36} \pm 1.46$ | $16.83 \pm 1.1$ |
| | 100 | $3.22 \pm 3.04$ | $17.19 \pm 1.31$ | $\mathbf{19.47} \pm 1.11$ | $19.12 \pm 1.48$ |
| *Ant* | 10 | $-24.25 \pm 20.86$ | $-33.98 \pm 19.87$ | $5.26 \pm 1.12$ | $\mathbf{6.04} \pm 1.72$ |
| | 25 | $-3.23 \pm 5.23$ | $-5.67 \pm 5.22$ | $11.27 \pm 1.57$ | $\mathbf{11.62} \pm 2.04$ |
| | 50 | $3.23 \pm 2.58$ | $0.55 \pm 3.38$ | $\mathbf{14.94} \pm 2.49$ | $13.07 \pm 4.24$ |
| | 100 | $7.72 \pm 1.14$ | $4.29 \pm 1.99$ | $\mathbf{18.17} \pm 4.7$ | $13.93 \pm 5.07$ |
| *Humanoid* | 10 | $-248.7 \pm 0.6$ | $\mathbf{-247.8} \pm 1.6$ | $-249.1 \pm 0.5$ | $-248.8 \pm 1.2$ |
| | 25 | $\mathbf{-247.8} \pm 0.5$ | $-247.9 \pm 0.7$ | $-248.6 \pm 0.3$ | $-248.8 \pm 0.1$ |
| | 50 | $\mathbf{-248.8} \pm 0.4$ | $-249 \pm 0.6$ | $-249.7 \pm 0.4$ | $-249.7 \pm 0.2$ |
| | 100 | $-249.5 \pm 0.3$ | $\mathbf{-249.4} \pm 0.6$ | $-249.8 \pm 0.3$ | $-249.8 \pm 0.1$ |

$p < 0.05$   $p < 0.01$   $p < 0.001$

A.5   INTRODUCTION TO NORMALIZING FLOWS

NFs are powerful non-parametric models that have demonstrated the ability to fit flexible multi-modal distributions (Tabak & Vanden-Eijnden, 2010; Tabak & Turner, 2013). These models achieve this by transforming a simple base continuous distribution, such as Gaussian or Beta, into a more complex one using the change of variable formula. By enabling scoring and sampling from the fitted distribution, NFs find application across various problem domains. Let $B$ represent the base distribution, a D-dimensional continuous random vector with $p_B(b)$ as its density function, and let $Y = g(B)$, where $g$ is an invertible function with an existing inverse $g^{-1}$, and both $g$ and $g^{-1}$ are differentiable. Leveraging the change of variable formula, we can express the distribution of $Y$ as follows:

$$p_Y(y) = p_B(g^{-1}(y))|\det(J(g^{-1}(y)))|, \tag{12}$$

where $J(\cdot)$ denotes the Jacobian, and $\det$ signifies the determinant. The first term on the right-hand side of Equation (12) governs the shape of the distribution, while $|\det(J(g^{-1}(y)))|$ normalizes it, ensuring the distribution integrates to one. Complex distributions can be effectively modeled by making $g(b)$ a learnable function with tunable parameters $\theta$, denoted as $g_\theta(b)$. However, it is essential to select $g$ carefully to guarantee its invertibility and differentiability. For examples of suitable choices, please refer to Papamakarios et al. (2021).

A.6   HYPOTHESIS TESTING DETAILS

We conducted Welch's t-tests to compare means $(\mu_i, \mu_j)$ between different estimators, as this test relaxes the assumption of equal variances compared to other hypothesis tests (Colas et al., 2019). The means for both KL and Bhatt were compared to each of the baseline methods: MC and random. To control the family-wise error rate (FWER), we performed a Holm-Bonferroni correction across each setting, environment, and acquisition batch. The calculated test statistics are presented in Tables 7, 6, and 8. It's important to note that test statistics for RMSE on PNEs are not reported as no values reached statistical significance.

Table 6: Test statistics for the log likelihood comparison for Nflows Base. Header denotes the two means being compared.

| Env | Acq. Batch | $(\mu_{rand}, \mu_{KL})$ | $(\mu_{MC}, \mu_{KL})$ | $(\mu_{rand}, \mu_{Bhatt})$ | $(\mu_{MC}, \mu_{Bhatt})$ |
|---|---|---|---|---|---|
| *hetero* | 10 | -4.25 | -0.35 | -3.79 | 0.89 |
| | 25 | -3.35 | 0.35 | -3.92 | 0.19 |
| | 50 | -3.71 | 0.56 | -4.73 | -0.05 |
| | 100 | -4.37 | -1.66 | -4.41 | -1.77 |
| *bimodal* | 10 | -7.54 | -1.45 | -7.54 | -1.43 |
| | 25 | -5.88 | 1.01 | -5.91 | -1.34 |
| | 50 | -4.25 | -0.30 | -4.26 | -0.46 |
| | 100 | -2.58 | 0.18 | -2.58 | 0.04 |
| *Pendulum* | 10 | -4.16 | 0.32 | -4.35 | -0.31 |
| | 25 | -5.28 | 0.72 | -5.24 | 0.98 |
| | 50 | -6.25 | 0.83 | -6.34 | 0.37 |
| | 100 | -12.68 | 1.71 | -12.83 | 1.84 |
| *Hopper* | 10 | -3.95 | 2.29 | -4.09 | 1.00 |
| | 25 | -13.55 | 3.00 | -13.61 | 1.69 |
| | 50 | -16.33 | -0.90 | -16.44 | -0.79 |
| | 100 | -16.66 | -0.62 | -17.75 | -0.85 |
| *Ant* | 10 | -4.64 | -4.56 | -4.76 | -4.64 |
| | 25 | -8.80 | -10.60 | -9.08 | -10.84 |
| | 50 | -9.79 | -11.13 | -8.36 | -9.82 |
| | 100 | -10.24 | -16.49 | -6.72 | -10.69 |
| *Humanoid* | 10 | -1.47 | -1.37 | -2.31 | -2.29 |
| | 25 | -0.17 | -0.30 | -1.47 | -1.48 |
| | 50 | 2.85 | 3.00 | -0.99 | -0.98 |
| | 100 | 6.38 | 6.24 | -2.72 | -2.80 |

Table 7: Test statistics for the RMSE comparison for Nflows Base. Header denotes the two means being compared.

| Env | Acq. Batch | $(\mu_{rand}, \mu_{KL})$ | $(\mu_{MC}, \mu_{KL})$ | $(\mu_{rand}, \mu_{Bhatt})$ | $(\mu_{MC}, \mu_{Bhatt})$ |
|-----|-----------|-----------|-----------|-----------|-----------|
| *hetero* | 10 | 3.33 | 0.75 | 0.63 | -1.20 |
| | 25 | 1.27 | -1.02 | 1.78 | 0.40 |
| | 50 | 2.49 | 0.47 | 1.71 | 0.03 |
| | 100 | 2.06 | 1.32 | 1.90 | 1.25 |
| *bimodal* | 10 | 2.11 | 0.14 | 2.12 | 0.18 |
| | 25 | 2.75 | 0.34 | 3.18 | 0.79 |
| | 50 | 2.31 | -0.21 | 2.34 | 0.08 |
| | 100 | 1.90 | -0.10 | 1.93 | 0.19 |
| *Pendulum* | 10 | 3.63 | 0.13 | 2.67 | -0.61 |
| | 25 | 4.17 | -1.18 | 4.45 | -1.36 |
| | 50 | 3.47 | -0.43 | 3.39 | -0.72 |
| | 100 | 7.83 | -1.84 | 7.64 | -2.17 |
| *Hopper* | 10 | 3.30 | -3.53 | 3.48 | -2.84 |
| | 25 | 10.11 | -3.20 | 12.50 | -1.40 |
| | 50 | 16.80 | 1.60 | 17.52 | 1.99 |
| | 100 | 10.33 | 2.81 | 10.10 | 1.87 |
| *Ant* | 10 | 1.14 | 3.36 | 0.49 | 2.72 |
| | 25 | -0.35 | 1.75 | 0.61 | 2.64 |
| | 50 | 2.10 | 5.29 | 0.59 | 3.44 |
| | 100 | 2.93 | 6.21 | 1.19 | 4.32 |
| *Humanoid* | 10 | 1.74 | 1.57 | -0.57 | -0.67 |
| | 25 | 3.90 | 4.35 | 1.27 | 1.65 |
| | 50 | 4.50 | 4.43 | 1.46 | 1.46 |
| | 100 | 5.71 | 5.73 | 2.89 | 2.72 |

Table 8: Test statistics for the log likelihood comparison for PNES. Header denotes the two means being compared.

| Env | Acq. Batch | $(\mu_{rand}, \mu_{KL})$ | $(\mu_{MC}, \mu_{KL})$ | $(\mu_{rand}, \mu_{Bhatt})$ | $(\mu_{MC}, \mu_{Bhatt})$ |
|---|---|---|---|---|---|
| *hetero* | 10 | -2.72 | 0.23 | -3.34 | 0.13 |
| | 25 | -4.95 | -0.85 | -4.90 | -1.04 |
| | 50 | -4.06 | 1.46 | -4.56 | 0.95 |
| | 100 | -5.41 | 0.83 | -4.71 | 2.34 |
| *bimodal* | 10 | -5.07 | -1.24 | -5.07 | -1.19 |
| | 25 | -7.67 | -1.13 | -7.67 | -1.17 |
| | 50 | -5.51 | -0.90 | -5.52 | -1.77 |
| | 100 | -2.75 | -0.83 | -2.75 | -0.92 |
| *Pendulum* | 10 | -4.46 | 2.23 | -4.46 | -0.04 |
| | 25 | -4.85 | 0.68 | -5.20 | 0.25 |
| | 50 | -7.50 | -1.55 | -6.38 | -0.29 |
| | 100 | -13.82 | 0.15 | -11.72 | -0.35 |
| *Hopper* | 10 | -4.21 | 1.62 | -4.19 | 1.66 |
| | 25 | -19.69 | -1.55 | -20.43 | -0.70 |
| | 50 | -18.56 | -3.68 | -17.94 | -1.57 |
| | 100 | -15.06 | -3.96 | -14.11 | -2.93 |
| *Ant* | 10 | -4.24 | -5.91 | -4.34 | -6.02 |
| | 25 | -7.97 | -9.32 | -7.94 | -9.25 |
| | 50 | -9.79 | -10.28 | -5.94 | -6.93 |
| | 100 | -6.48 | -8.16 | -3.59 | -5.31 |
| *Humanoid* | 10 | 9.15 | 5.92 | 4.52 | 4.38 |
| | 25 | 8.99 | 6.96 | 11.11 | 8.29 |
| | 50 | 5.08 | 3.48 | 5.93 | 3.89 |
| | 100 | 2.41 | 1.82 | 3.35 | 2.21 |

