# OpenReview forum: "Escaping the Sample Trap: Fast and Accurate Epistemic Uncertainty Estimation with Pairwise-Distance Estimators"
_ICLR.cc/2024/Conference — Submitted to ICLR 2024_

### Official Review · Reviewer_H6Rh · 2023-10-30

**Soundness:** 3 good
**Presentation:** 3 good
**Contribution:** 3 good
**Rating:** 6
**Confidence:** 3

**Summary:**

The paper is concerned with developing a new estimator for epistemic uncertainty. Epistemic uncertainty can be used as a criterion to guide active learning in the ensemble learning setting where there are multiple base learners (instantiated as neural networks). The estimator is combined with nflows base, which is a method to learn base learners. The new estimators can avoid Monte Carlo sampling. They speed up learning and can be applied to problems of larger scale.  Experiments are carried out on a variety of datasets. The proposed estimators are compared with the conventional Monte Carlo estimator.

**Strengths:**

* originality: the proposed estimator is based on Kolchinsky & Tracey (2017). The application is new.
* Quality: the work is of good quality. the proposed estimator has advantages over existing ones such as closed form, theoretical tie to epistemic uncertainty, efficiency, and scalability. The estimation quality is also good compared to the Monte Carlo based estimator.
* clarity: presentation is clear.
* significance: The proposed method is of good significance as it proposed a new method that is better than the existing one for this specific type of problem.

**Weaknesses:**

* Given Kolchinsky & Tracey (2017), the contribution of this work is somewhat incremental.
* Only NFLows Base is considered as the base learner method for ensemble learning. It would be nice to see if the proposed estimator can also be applied to other scenarios.
* The application scenario that the proposed method applied to also appears to be quite niche.

**Questions:**

Some suggestions on the writing of the paper:
* eq 1, needs to specify pi_j  to be between 0 and 1.
* the use of \theta is somewhat confusing. It sometimes can refer to the parameters of the entire ensemble but other times it refers to the parameters of the base learner (e.g. eq1 vs the eq right before eq3).
* grammar: "in addition to the improving the lower bound"
* would be nice to list the environment dimension and other related statistics in Table 1 for readability.
* Figure 2:  show the names of the x-axis and y-axis.
* would be nice to show how eq3 is derived.

Other questions:
* any suggestions on whether one should use the lower bound or the upper bound estimator?
* How does one use eq 2 to carry out active learning? does it imply that y needs to be known for all the data points in the dataset? A pseudocode list for the entire procedure may help to explain things better.

---

> ### Author Response · Authors · 2023-11-21
>
> Thank you for the review.
>
> &nbsp;
>
> Additional Base Learners:
> &nbsp;
>
> We do consider Probabilistic Network Ensembles as an additional base learner in the Appendix.
>
> &nbsp;
>
> Writing Suggestions:
> &nbsp;
>
> Thank you, these have been implemented and are reflected in the new version of the paper.
>
> &nbsp;
>
> Thank you for your thoughtful comments and the time you invested in reviewing our paper.

---

### Official Review · Reviewer_rbCP · 2023-11-01

**Soundness:** 3 good
**Presentation:** 3 good
**Contribution:** 3 good
**Rating:** 5
**Confidence:** 5

**Summary:**

This paper proposes an active learning algorithm using the pairwise distances of densities (probabilities) to approximate the mutual information-like metric such as $H(y \mid x) – H(y \mid x, \theta ).$ The basic framework start from the inequalities of $H( y \mid x , \theta) \le H( y \mid x) \le H( y, \theta \mid x).$ The proposed approximation for $H( y \mid x) $ uses the distance between $p_i( y \mid x, \theta_i ) $ and $p_j ( y \mid x, \theta_j)$ where two distribution are from the uncertainty of $\theta.$ The distance between two densities utilizes the KL divergence and Bhattacharyya distance. Furthermore, normalizing flow is used to generate the ensembles for $p( y \mid x, \theta_i)$. Experiments results are promising. Relatively lower or comparable computation cost is required compared MC method. High receptive power for the unlabeled samples and high accuracy in high-dimensional data.

**Strengths:**

The proposed algorithm is clever, and it have the theoretical validation although the approximation cannot be tight in some cases. In the computational cost, the advantages are obvious. Considering the complexity in Bayesian approach, the proposed algorithm is superior to MC, not depending on variational inference. Also, the performance is superior to the MC approach.

**Weaknesses:**

The baseline-algorithms are not sufficient, Random (query), BADGE (Ash et al., 2019) and BatchBALD (Kirsch et al., 2019) can be other base-line algorithms. The author(s) claim that the proposed algorithm can be strong for the large size of queries. Therefore, I want to see the compared results with algorithms considering the diversity. If possible, you can use the multiple samples for $p(y \mid x_1, \cdots, x_d, \theta_i)$. More experiments and presentation can improve the paper, including the various acquisition sizes and trace plot of test accuracy and so on.

**Questions:**

Q1: Is the notation of P_{b \mid b} in 31 lines on page 5 right?
Q2: Please clarify what is f(a_t, s_t) = s_{t+1} in a detailed manner.
Q3: Number of samples in Figure 3: what’s the meaning, probably, size of unlabeled samples at initial steps, right?

**Details Of Ethics Concerns:**

None.

---

> ### Author Response · Authors · 2023-11-21
>
> Thank you for the review.
>
> &nbsp;
>
> Baselines:
> &nbsp;
>
> Random query is used as a baseline, as detailed in Table 1-5 and depicted in Figures 7, 8, 10, and 13. Our primary contribution lies in presenting a more precise estimator of epistemic uncertainty in higher dimensions. Consequently, we have adopted BALD (Houlsby et al. 2011) as a baseline in our experiments. BADGE (Ash et al., 2019) computes the gradient of the last layer outputs and employs k-means++ on these gradients to select data points for acquisition. In their original implementation, they exclusively focused on classification tasks with gradients of size 256x10. In our experiments, we face the challenge of computing gradients of size 256x257x2 and subsequently applying k-means++. However, k-means++ encounters difficulties in tasks with such elevated dimensionality. Moreover, BatchBALD necessitates the estimation of $p(y_1,...,y_n)$, a task feasible in classification but poses a barrier in high-dimensional regression output where $y_i$ represents a continuous output in a high-dimensional space.
>
> &nbsp;
>
> Plot Test Accuracy:
> &nbsp;
>
> Please be aware that our work addresses regression tasks, in contrast to the majority of previous studies that primarily concentrate on classification as in your suggested baselines. Consequently, accuracy is not an applicable metric in our context. This distinction is emphasized throughout the paper, including in the Related Works section, where we state, “Our experiments encompassed tasks where the output spans continuous distributions for 1 to 257 dimensions, as opposed to the aforementioned methods that primarily address classification problems with a 1D categorical output.”  For detailed results, we provide RMSE and Log Likelihood metrics in the Appendix.
>
> &nbsp;
>
> Q1:
> &nbsp;
>
> Thank you for pointing this out, we have addressed this small typo in the text.
>
> Q2:
> &nbsp;
>
> The function $f$ denotes a transition function: given a state and action, denoted as $s_t$ and $a_t$ respectively, it produces the subsequent state $s_{t+1}$. In our scenario, these functions are intricate and nonlinear due to our consideration of robot dynamics. In the paper, we explicitly note this as a dynamics model, stating, "the dynamics model for each environment was modeled, $f_{\theta} (s_t, a_t) = s_{t+1}$."
>
> &nbsp;
>
> Q3:
> &nbsp;
>
> Monte Carlo (MC) estimators necessitate the drawing of a certain number of samples to provide accurate estimates; typically, fewer samples lead to poorer results. Figure 3 illustrates this phenomena within a specific environment. To enhance clarity, we have modified the text in the paper accordingly:
> &nbsp;
>
> To conduct a more in-depth analysis of our proposed method, we compared PaiDEs to MC estimators with a varying sample size $per$ $x_i$ in the Hopper-v2 environment. We expected that MC estimators would perform on par with PaiDEs with a sufficient number of samples. However, as illustrated in Figure 3,MC estimators fell short of achieving the same level of performance as PaiDEs in this particular scenario. This suggests that, taking into consideration hardware constraints, PaiDEs begin to outperform MC estimators when dealing with 11 dimensional outputs.
>
> &nbsp;
>
> Thank you for your thoughtful comments and the time you invested in reviewing our paper. We have carefully considered your concerns and believe that our responses adequately address them. We kindly request you to consider increasing your score, as it would greatly enhance the likelihood of our paper being accepted.

---

> > ### Comment · Reviewer_rbCP · 2023-12-01
> > **Response to Rebuttal**
> >
> > Thanks for your answer. Some issues are resolved. However, we cannot ensure that we have sufficient baselines for the comparison. The authors focus on the regression task, which explains the limitations of the experiments. It can be another limitation since the classification tasks are crucial in active learning.

---

### Official Review · Reviewer_ts3p · 2023-11-08

**Soundness:** 3 good
**Presentation:** 3 good
**Contribution:** 2 fair
**Rating:** 5
**Confidence:** 5

**Summary:**

The authors propose an ensemble method for uncertainty quantification, specifically focusing on mutual information as a measure of epistemic uncertainty. Making use of approximations of (conditional) entropy in terms of pairwise distance measures on distributions, they propose a method for approximating mutual information in terms of a sum of pairwise distances between ensemble members. If these distances can be computed in closed form, the whole approach becomes very efficient, especially in comparison to standard sampling-based methods. The authors also present promising experimental results.

**Strengths:**

Very interesting and timely topic.
The idea of pairwise approximation is quite appealing and theoretically justified.
The experiments are well conducted, and the results are promising.

**Weaknesses:**

Some parts of the paper are very concise and hard to follow. In particular, this is true for Sections 5 and 6. The combination of PaiDEs and Nflows Base is not very clear to me.

**Questions:**

How exactly is Nflows Base used to create an ensemble? This didn’t become very clear in the paper, but I think it is an important point, because the variability of the ensemble members eventually determines the epistemic uncertainty. In standard ensemble learning, variability is normally enforced through randomisation, e.g., by resampling the training data or randomly initialising weights in a neural network. Here, the risk is that different randomisations may lead to different variability, making (epistemic) uncertainty arbitrary to some extent. So how is the variability produced by the NF ensemble controlled, and in which sense is it “meaningful” or “natural”?

Another point that puzzled me is the connection between the base distributions and the output distributions produced by Nflows Base. This connection is illustrated by Figure 1 (I was wondering why this figure is shown right in the beginning, although it is only referenced on page 6). What I suspect is that the pairwise distances between the output distributions is the same as the pairwise distances between the base distributions. This seems to be a key point of the approach. However, it is not made very explicit in the paper. Moreover, why exactly are the distances the same? The distributions actually look quite different. Shouldn’t this be shown in a formal way?

Another question in this regard, also related to the first one: Where exactly are the base distributions combing from in the first place. Again, as these distributions seem to determine the epistemic uncertainty, this looks to me like an important question.

---

> ### Author Response · Authors · 2023-11-21
>
> Thank you for the review.
>
> &nbsp;
>
> PaiDEs and Nflows Base:
> &nbsp;
>
> Please be aware that an introduction to Normalizing Flows is available in the appendix (A.5). To provide more clarity, we have modified the text in the paper as follows:
> &nbsp;
>
> Berry & Meger (2023) demonstrate that estimating Equation (2) in the base distribution is equivalent to estimating it in the output distribution. Consequently, by combining Nflows Base and PaiDEs, we construct an expressive non-parametric model capable of capturing intricate aleatoric uncertainty in the output distribution while efficiently estimating epistemic uncertainty in the base distribution. Unlike previously proposed methods, we are able to estimate epistemic uncertainty without taking a single sample. Figure 1 shows an example of the distributional pairs that need to be considered in order to estimate epistemic uncertainty for an Nflows Base model with 3 components
>
> &nbsp;
>
> Nflows Base and Base Distribution:
> &nbsp;
>
> When fitting a normalizing flow the base distribution is needed to create the flow. There is an introduction to Normalizing flows in the appendix but for more details refer to Papamakarios et al. (2021). To address this confusion we have changed the text to:
> &nbsp;
>
>  Nflows Base creates an ensemble in the base distribution,
> \begin{align*}
> 	p_{y|x,\theta}(y|x,\theta) &= f_{\theta_j}(y,x) = p_{b|x,\theta}(g_{\theta}^{-1}(y,x))|\det(J(g_{\theta}^{-1}(y,x)))|,
> \end{align*}
> where $p_{b|x,\theta}(b|x,\theta)=N(\mu_{\theta,x}, \Sigma_{\theta,x})$, $\mu_{\theta,x}$ and $\Sigma_{\theta,x}$ denote the mean and covariance conditioned on both $x$ and $\theta$. These parameters are modeled using a neural network with fixed dropout masks to establish an ensemble and ensemble diversity is created by randomization and bootstrapping.
>
> &nbsp;
>
> Thank you for your thoughtful comments and the time you invested in reviewing our paper. We have carefully considered your concerns and believe that our responses adequately address them. We kindly request you to consider increasing your score, as it would greatly enhance the likelihood of our paper being accepted.

---

### Official Review · Reviewer_3c49 · 2023-11-10

**Soundness:** 3 good
**Presentation:** 3 good
**Contribution:** 2 fair
**Rating:** 3
**Confidence:** 4

**Summary:**

The paper introduces Pairwise-Distance Estimators (PaiDEs) as an efficient non-sampling approach to estimate epistemic uncertainty for ensemble models. PaiDEs utilize pairwise distances between ensemble component distributions to estimate entropy and mutual information, avoiding expensive Monte Carlo sampling. This enables faster uncertainty quantification that scales more favorably to high dimensions. Experiments on active learning tasks using MuJoCo environments demonstrate that the approach can accurately quantify epistemic uncertainty. Key benefits of the proposed PaiDE technique include removing dependence on sampling and improved computational performance compared to sampling-based approaches. Normalizing flow ensembles are employed to flexibly model aleatoric uncertainty while PaiDEs leverage their tractable base distributions.

**Strengths:**

Originality:

Proposes a novel non-sampling approach to estimate epistemic uncertainty, which is an open problem
Leverages pairwise distances in a creative way to avoid sampling limitations

Quality:
Good mathematical grounding in information theory and probability
Principled design of PaiDE estimators with entropy bounds
Careful methodology and experiment design

Clarity:
Clearly explains limitations of sampling-based approaches
Intuitive descriptions and visualizations of the methodology
Well-structured layout and organization

**Weaknesses:**

No comparison to other state-of-the-art uncertainty methods - It will add credibility to show benefits over existing approaches. For eg. comparing against Deterministic UQ Methods - ND-UQ (van Amersfoort et al., 2020), Active Learning Methods - BALD (Houlsby et al., 2011), MC Dropout - Estimate epistemic uncertainty using dropout at test time as a Bayesian approximation (Gal and Ghahramani, 2016), Ensembles - Compare to other ensemble approaches like Deep Ensembles (Lakshminarayanan et al., 2017).

OOD detection are not evaluated - This is a missed opportunity to demonstrate more utility of accurate epistemic uncertainty estimation.

The empirical evaluation is quite limited in scope due the first point - lacks sufficient evidence that method works in complex real domains.

Theoretical understanding is completely lacking, because the experiments are simple and toy scenarios - analysis of estimator bias, convergence, optimality properties are important.

Scaling behavior as ensemble size increases is unclear - needs more analysis on this.
it would be useful to see a study on how the performance of the proposed PaiDE method scales as the number of components in the ensemble increases. For example, the experiments in the paper use a small ensemble of 5 models. It would be interesting to additionally evaluate with larger ensembles of 10, 50+ components.
Key questions that could be investigated:
Does the accuracy of epistemic uncertainty estimates improve with more components?
Is there a point of diminishing returns where more components do not help?
How does this compare to scaling behavior of sampling-based MC approaches?

**Questions:**

Please see the questions

---

> ### Author Response · Authors · 2023-11-21
>
> Thank you for the review.
>
> &nbsp;
>
> Baselines:
> &nbsp;
>
> Please be aware that BALD (Houlsby et al., 2011) selects inputs, $x$'s, that maximize the mutual information criterion as described in Equation (2). This framework is applied in all experiments of our paper as stated in the paper though to make this more clear we have added the following text:
> &nbsp;
>
> We select data points that maximize Equation (2) as in Bayesian Active Learning by Disagreement (BALD) to improve the model’s performance (Houlsby et al., 2011)
> &nbsp;
>
> Our proposal introduces a novel, more efficient method to estimate this criterion, particularly beneficial for high-dimensional outputs, while performing similarly in lower dimensions. It's important to note that Deterministic UQ Methods (ND-UQ) by van Amersfoort et al. (2020) are only applicable to classification tasks, which are not within the scope of our work. Deep Ensembles, such as those by Lakshminarayanan et al. (2017), are discussed in the appendix (A.4). MC Dropout has some challenges at estimating epistemic uncertainty for regression tasks (Berry and Meger 2023).
>
> &nbsp;
>
> OOD:
> &nbsp;
>
> The performance in active learning suggests that the method can identify data points that are underrepresented in the training set, providing insight into how our approach may perform in detecting out-of-distribution (OOD) samples. There is a considerable overlap between epistemic uncertainty estimation and OOD detection, with many of our results demonstrating the method's proficiency in identifying novel inputs.
>
> &nbsp;
>
> Limited Experiments:
> &nbsp;
>
> Two of the baselines, claimed missing, are actually included in the paper. One of them is inapplicable to our specific setting, and the last baseline demonstrates poor performance in our context. Please refer to the earlier response regarding the Baselines for more details. It's important to note that Mujoco simulates realistic and complex robot dynamics, with significantly higher dimensions compared to datasets in prior papers. Consequently, our experiments tackle a more challenging problem than those addressed in the existing literature, and this is the specific gap we aim to fill.
>
> &nbsp;
>
> Theoretical Analysis:
> &nbsp;
>
> The bias of the proposed methods is examined in Figure 6 and 11 in the Appendix and mentioned in Section 7.4 of the paper. Furthermore, we offer a theoretical analysis illustrating when our proposed method may encounter challenges based on the number of components, as illustrated in Figure 4.
>
> &nbsp;
>
> Scaling Behavior:
> &nbsp;
>
> Scaling behavior is investigated in Figure 4. The estimation of epistemic uncertainty would accurately scale with the number of components, though implementing the experiments you propose would demand a significant computational resource. These resources are typically only available to large technology companies.
>
> &nbsp;
>
> Thank you for your thoughtful comments and the time you invested in reviewing our paper. We have carefully considered your concerns and believe that our responses adequately address them. We kindly request you to consider increasing your score, as it would greatly enhance the likelihood of our paper being accepted.

---

### Meta-Review · Area_Chair_mLL8 · 2023-12-09

**Metareview:**

Although the reviewers raise a number of critical points in their original reports, there is agreement that the paper holds promise, and the authors' idea of using pairwise distance estimators to avoid the need for sampling in uncertainty quantification looks quite intriguing. The authors also showed a high level of commitment during the rebuttal phase and did their best to respond to the comments and to improve the submission. This was appreciated and positively acknowledged by all. In the discussion between authors and reviewers, some critical points could be resolved and some questions clarified. Other points remained open and were critically reconsidered in the subsequent internal discussion. In particular, the reviewers are still not very convinced by the empirical validation and are missing other baselines that should be included and compared with, as well as other practical settings. The theoretical contribution of the paper may appear somewhat limited in light of the prior work the authors are building on. Last but not least, the presentation could still be improved.

**Justification For Why Not Higher Score:**

Empirical validation not convincing, missing baselines and other practical settings. Theoretical contribution somewhat limited. Scope to improve presentation.

**Justification For Why Not Lower Score:**

N/A

---

### Decision · Program_Chairs · 2024-01-16

Reject